# Private Stochastic Convex Optimization with Heavy Tails: Near-Optimality from Simple Reductions

**Hilal Asi**
Apple Inc.
hilal.asi94@gmail.com

**Daogao Liu** *
University of Washington
liudaogao@gmail.com

**Kevin Tian**
University of Texas at Austin
kjtian@cs.utexas.edu

## Abstract

We study the problem of differentially private stochastic convex optimization (DP-SCO) with heavy-tailed gradients, where we assume a $k^{\text{th}}$-moment bound on the Lipschitz constants of sample functions, rather than a uniform bound. We propose a new reduction-based approach that enables us to obtain the first optimal rates (up to logarithmic factors) in the heavy-tailed setting, achieving error $G_2 \cdot \frac{1}{\sqrt{n}} + G_k \cdot (\frac{\sqrt{d}}{n\varepsilon})^{1-\frac{1}{k}}$ under $(\varepsilon, \delta)$-approximate differential privacy, up to a mild $\mathrm{polylog}(\frac{1}{\delta})$ factor, where $G_2^2$ and $G_k^k$ are the $2^{\text{nd}}$ and $k^{\text{th}}$ moment bounds on sample Lipschitz constants, nearly-matching a lower bound of [LR23]. We further give a suite of private algorithms in the heavy-tailed setting which improve upon our basic result under additional assumptions, including an optimal algorithm under a known-Lipschitz constant assumption, a near-linear time algorithm for smooth functions, and an optimal linear time algorithm for smooth generalized linear models.

## 1 Introduction

Differentially private stochastic convex optimization (DP-SCO), where an algorithm aims to minimize a population loss given samples from a distribution, is a fundamental problem in statistics and machine learning. In this problem, given $n$ samples from a distribution $\mathcal{P}$ over a sample space $\mathcal{S}$, our goal is to privately find an approximate minimizer $\hat{x} \in \mathcal{X} \subset \mathbb{R}^d$ for the population loss

$$F_{\mathcal{P}}(x) := \mathbb{E}_{s \sim \mathcal{P}} \left[ f(x; s) \right],$$

where $f(\cdot; s)$ is a convex function for all $s \in \mathcal{S}$. The quality of an algorithm is measured by the excess population loss of its output $\hat{x}$, that is $F_{\mathcal{P}}(\hat{x}) - \min_{x^\star \in \mathcal{X}} F_{\mathcal{P}}(x^\star)$.

Extensive research efforts have been devoted to DP-SCO, resulting in important progress over the past few years [BFTT19, FKT20, AFKT21, BGN21, ALD21, KLL21]. In an important milestone, [BFTT19] developed optimal algorithms (in terms of the excess population loss) for DP-SCO under a uniform Lipschitz assumption (i.e., where every $f(\cdot; s)$ is assumed to have the same Lipschitz bound), and [FKT20] followed this result with efficient and optimal algorithms that run in linear time for smooth functions. DP-SCO has also been explored in other notable settings, including developing faster algorithms for non-smooth settings [AFKT21, KLL21, CJJ+23], different geometries imposed on the solution space [AFKT21, BGN21, GLL+23], and different notions of privacy [ALD21].

Most existing results in DP-SCO are based on the assumption that the function $f(\cdot; s)$ is uniformly $G$-Lipschitz for all $s \in \mathcal{S}$. This assumption is convenient for private algorithm design, because it allows us to straightforwardly bound the *sensitivity* of iterates of private algorithms, i.e., how far a pair of iterates defined via algorithms induced by neighboring datasets drift apart. Under the uniform Lipschitz assumption, the DP-SCO problem is relatively well-understood, as optimal and efficient

---

*Part of this work was done while interning at Apple.

algorithms exist (sometimes requiring additional regularity assumptions) [BFTT19, FKT20].[2] State-of-the-art SCO algorithms satisfying $(\varepsilon, \delta)$-differential privacy (Definition 1) in the uniform Lipschitz setting result in excess population loss

$$GD\left(\frac{1}{\sqrt{n}} + \frac{\sqrt{d\log(\frac{1}{\delta})}}{\varepsilon n}\right), \tag{1}$$

where $D$ is the diameter of $\mathcal{X}$. However, the assumption of uniformly $G$-Lipschitz gradients is strong, and may be violated in real-life applications where the distribution in question has heavy tails (see e.g. discussion in [ACG$^+$16]). As a simple motivating example, consider mean estimation, where each $f(\cdot; s) = \frac{1}{2}\|\cdot - s\|^2$, so the minimizer of $F_{\mathcal{P}}$ is the population mean. The uniform Lipschitz requirement amounts to $\mathcal{P}$ having a bounded support over $\mathcal{X}$, whereas an algorithm that can handle heavy tails only posits the weaker assumption that $\mathcal{P}$ has bounded $k$-th moments. However, as pointed out by [WXDX20], many real-world datasets [MM97, BDFS07, IIW15], especially those from biomedicine and finance, are usually unbounded or even heavy-tailed. As a result, existing algorithms for DP-SCO may have overly pessimistic performance bounds when $G$ is large or even unbounded, necessitating the search for new private algorithms handling heavy-tailed gradients.

Motivated by this weakness of existing DP-SCO analyses, several papers studied the problem of DP-SCO with heavy-tailed gradients [WXDX20, ADF$^+$21, KLZ22, LR23], formally defined in Definition 4. Rather than assuming uniformly Lipschitz gradients, this line of work builds on the more realistic assumption that the norm of the gradients has bounded $k^{\text{th}}$-moments. In particular, [ADF$^+$21] studied heavy-tailed private optimization for the related empirical loss, while [WXDX20] initiated an analogous study for the population loss. More recently, [KLZ22, LR23] also proposed algorithms to solve the heavy-tailed DP-SCO problem based on clipped stochastic gradient methods.

Despite the significant progress made in addressing heavy-tailed DP-SCO, it remains notably less understood compared to the uniformly Lipschitz setting. As a benchmark, under a notion called $\rho$-concentrated differential privacy (CDP, see Definition 3), which translates to $(\varepsilon, \delta)$-DP for $\rho \approx \varepsilon^2 \log^{-1}(\frac{1}{\delta})$, [LR23] established that the best excess population loss achievable scales as

$$G_2 D \cdot \frac{1}{\sqrt{n}} + G_k D \cdot \left(\frac{\sqrt{d}}{n\sqrt{\rho}}\right)^{1-\frac{1}{k}}, \tag{2}$$

where $G_j^j$ is the $j^{\text{th}}$ moment bound on the Lipschitz constant of sampled functions, see Definition 4. Note that as $k \to \infty$, the rate in (2) recovers the uniform Lipschitz rate in (1).

Unfortunately, existing works on heavy-tailed DP-SCO assume stringent conditions on problem parameters and are suboptimal in the general case. For example, [KLZ22] requires the loss functions to be uniformly smooth with various parameter bounds in order to guarantee optimal rates, while the recent work [LR23] obtains a suboptimal rate scaling as[3] $G_2 D \cdot \frac{1}{\sqrt{n}} + G_k D \cdot (\frac{\sqrt{d}}{n\sqrt{\rho}})^{1-\frac{2}{k}}$, which is worse than (2) by polynomial factors in the dimension for any constant $k$.

## 1.1 Our contributions

Motivated by the suboptimality of existing results for heavy-tailed DP-SCO, we develop the first algorithm for this problem, which achieves the optimal rate (2) up to logarithmic factors with no additional assumptions. Along the way, we give several simple reduction-based tools for overcoming technical barriers encountered by prior works. To state our results (deferring a formal problem statement to Definition 1), we assume that for some $k \geq 2$ and all $j \in [k]$, we have

$$\mathbb{E}_{s\sim\mathcal{P}}\left[\max_{x\in\mathcal{X}}\|\nabla f(x; s)\|^j\right] \leq G_j^j.$$

Our results hold in several settings and are based on different reductions which allow us to apply strategies for DP-SCO from the uniform Lipschitz setting.

---

[2]One notable exception is the lack of linear-time algorithms in the non-smooth setting.

[3]The rate in [LR23] is stated slightly differently (see their Theorem 6), as they parameterize their error bound via $G_{2k}$ despite assuming only $k$ bounded moments. However, under the assumption that $G_{2k}$ is finite (so the [LR23] result is usable), the optimal rate scales as in (2) where $k$ is replaced with $2k$, leaving a polynomial gap.

**Near-optimal rates for heavy-tailed DP-SCO (Section 3).** We design an algorithm for the $k$-heavy-tailed DP-SCO problem, which satisfies $\rho$-CDP[4] and attains near-optimal excess loss

$$G_2 D \cdot \sqrt{\frac{\log\left(\frac{1}{\delta}\right)}{n}} + G_k D \cdot \left(\frac{\sqrt{d}\log\left(\frac{1}{\delta}\right)}{n\sqrt{\rho}}\right)^{1-\frac{1}{k}}. \tag{3}$$

This matches the lower bounds recently proved by [KLZ22, LR23] for $\rho$-concentrated DP algorithms up to logarithmic factors, stated in (2). Standard conversions from CDP to $(\varepsilon, \delta)$-DP imply that our algorithm also obtains loss $\approx G_2 D \cdot \sqrt{\frac{1}{n}} + G_k D \cdot (\frac{\sqrt{d\log^3(1/\delta)}}{n\varepsilon})^{1-\frac{1}{k}}$ under this parameterization. We note that our bound (3) holds with high probability $\geq 1 - \delta$, whereas the lower bound (2) is for an error which holds only in expectation (see Theorem 13, [LR23]). Our lossiness in (3) is due to a natural sample-splitting strategy used to boost our failure probability, and we conjecture that (3) may be optimal in the high-probability error bound regime.

As in [LR23], to establish our result we begin by deriving utility guarantees for a clipped stochastic gradient descent subroutine on an empirical loss, where clipping ensures privacy but induces bias, parameterized by a dataset-dependent quantity $b_{\mathcal{D}}^2$ defined in (26). We give a standard analysis of this subroutine in Proposition 1, a variant of which (with slightly different parameterizations) also appeared as Lemma 27, [LR23]. However, the key technical barrier encountered by the [LR23] analysis, when converting to population risk, was bounding $\mathbb{E}b_{\mathcal{D}}^2$ over the sampled dataset, which naïvely depends on the $2k^{\text{th}}$ moment of gradients. This either incurs an overhead depending on $G_{2k}$, or in the absence of such a bound (which is not given under the problem statement), leads to the aforementioned suboptimal rate in [LR23] losing a factor of $(\frac{\sqrt{d}}{n\sqrt{\rho}})^{\frac{1}{k}}$ in the utility. We give a further discussion of natural strategies and barriers towards directly bounding $\mathbb{E}b_{\mathcal{D}}^2$ in Appendix G.

Where we depart from the strategy of [LR23] is in the use of a new *population-level localization* framework we design (see Algorithm 2), inspired by similar localization techniques in prior work [FKT20] (discussed in more detail in Section 1.2). This strategy allows us to use constant-success probability bounds on the quantity $b_{\mathcal{D}}$ (which also bound $b_{\mathcal{D}}^2$), which are easy to achieve depending only on $G_k$ rather than $G_{2k}$ via Markov's inequality. This bypasses the need in [LR23] for bounding $\mathbb{E}b_{\mathcal{D}}^2$. The motivation for population-level localization is that we wish to aggregate empirical solutions to multiple datasets, some of which have small $b_{\mathcal{D}}$, and others which do not. However, each dataset has a different empirical minimizer, so it is unclear how to argue about convergence if we apply the empirical localization. Instead, we aggregate solutions close to the population-level minimizer and share them across datasets via a simple geometric aggregation technique, showing that it suffices for a constant fraction of datasets to have this desirable property for us to carry out our population-level localization argument. We formally state our main result achieving the rate (3) as Theorem 1.

Interestingly, as a straightforward corollary of our new localization framework, we achieve a tight rate for high-probability stochastic convex optimization under a bounded-variance gradient estimator parameterization, perhaps the most well-studied formulation of SCO. To our knowledge, this result was only first achieved very recently by [CH24].[5] However, we find it a promising proof-of-concept that our new framework directly yields the same result. For completeness, we include a derivation in Appendix E (see Theorem 5) as a demonstration of the utility of our framework.

**Optimal rates with known Lipschitz constants (Appendix B).** We next consider the *known Lipschitz* setting, where each sample function $f(\cdot; s)$ arrives with a value $\overline{L}_s$ which is an overestimate of its Lipschitz constant, such that $\mathbb{E}\overline{L}_s^j$ is bounded for all $j \in [k]$ (see Assumption 2). As motivation, consider the problem of learning a generalized linear model (GLM), where $f(\cdot; s) = \sigma(\langle \cdot, s \rangle)$ for a known convex activation function $\sigma$. Typically, the Lipschitz constant for $f(\cdot; s)$ is simply the Lipschitz constant of $\sigma$ times $\|s\|$, which can be straightforwardly calculated. Thus, for GLMs, our known Lipschitz heavy-tailed assumption amounts to moment bounds on the distribution $\mathcal{P}$.

---

[4]We state the privacy guarantee of most of our results, save our algorithm in Appendix C which employs the sparse vector technique of [DNR+09, DR14], in terms of CDP, for simpler comparison to the lower bound (2).

[5]We mention that an alternative route to obtaining a near-optimal high-probability rate was given slightly earlier in [SZ23], but lost a polylogarithmic factor in the failure probability. We also wish to acknowledge that in an independent and concurrent work [JST24] involving the third author, the authors slightly sharpened and generalized the result of [SZ23], which inspired us to consider this application of our population-level localization framework.

Our second result, Theorem 2, shows a natural strategy obtains optimal rates in this known Lipschitz setting, eliminating logarithmic factors from Theorem 1. As mentioned previously, this result holds for the important family of GLMs. Our algorithm is based on a straightforward reduction to the uniformly Lipschitz setting: after simply iterating over the input samples, and replacing samples whose Lipschitz constant exceeds a given threshold with a new dummy sample, we show existing Lipschitz DP-SCO algorithms then obtain the optimal heavy-tailed excess population loss (2). Despite the simplicity of this result, to the best of our knowledge, it was not previously known.

**Efficient algorithms for smooth functions (Appendices C and D).** Finally, we propose algorithms with improved query efficiency for general smooth functions or smooth GLMs, with moderate smoothness bounds. Our strategy is to analyze the stability of clipped-DP-SGD in the smooth heavy-tailed setting, and use localization-based reductions to transform a stable algorithm into a private one [FKT20]. This results in linear-time algorithms for the smooth case with near-optimal rates. In order to prove the privacy of our smooth, heavy-tailed algorithm, we analyze a careful interplay of our clipped stochastic gradient method with the sparse vector technique (SVT) [DNR+09, DR14]. At a high level, our use of SVT comes from the fact that under clipping, smooth gradient steps no longer enjoy the type of contraction guarantees applicable in the uniform Lipschitz setting (see Fact 3), so we must take care to not clip too often. The SVT is then used to ensure privacy of our count of how many clipping operations were used. In Appendix F, we provide a simple counterexample showing that the noncontractiveness of contractive steps after applying clipping is inherent. Our general smooth heavy-tailed DP-SCO result is stated as Theorem 3.

We believe the use of SVT within an optimization algorithm to ensure privacy may be of independent interest, as it is one of few such instances that have appeared in the private optimization literature to our knowledge; it is inspired by a simpler application of this technique carried out in [AL24].

On the other hand, we make the simple observation that for GLMs, clipping cannot make a contractive gradient step noncontractive, by taking advantage of the fact that the derivative of $f(x; s) = \sigma(\langle x, s \rangle)$ is a multiple of $s$ for any $x \in \mathcal{X}$ (see Lemma 14). We use this observation to give a straightforward adaptation of the smooth algorithm in [FKT20] to the heavy-tailed setting, proving Theorem 4, which attains both a linear gradient query complexity and the optimal rate (2).

## 1.2 Prior work

The best-known rates for heavy-tailed DP-SCO were recently achieved by [KLZ22, LR23]. As discussed previously, their results do not provide the same optimality guarantees as our Theorem 1. The rate achieved by [LR23] is polynomially worse than the optimal loss (2) for any constant $k$. On the other hand, the work of [KLZ22] uses a different assumption on the gradients than Assumption 1, which is arguably more nonstandard: in particular, they require that the $k^{\text{th}}$-order central moments of each coordinate $\nabla_j f(x; s)$ is bounded. Moreover, their algorithms require each sample function $f(\cdot; s)$ to be $\beta$-smooth, and the final rates have a strong dependence on the condition number $\kappa = \frac{\beta}{\lambda}$ where $\lambda$ is the strong convexity parameter (see Appendix C in [LR23] for additional discussion).

Our result in the heavy-tailed setting assuming $\beta$-smoothness of sample functions, Theorem 3, is most directly related to Theorem 15 of [LR23]. These two results respectively require

$$\beta = O\left(\frac{G_k}{D} \cdot \varepsilon^{1.5} \sqrt{\frac{n}{d}}\right) \text{ and } \beta = O\left(\frac{G_k}{D} \cdot \left(\frac{d^5}{\varepsilon n}\right)^{\frac{1}{18}}\right),$$

omitting logarithmic factors in our bound for simplicity, to obtain near-optimal rates. These regimes are different and not generally comparable. However, we find it potentially useful that our upper bound on $\beta$ grows as more samples are taken, whereas the [LR23] bound degrades with larger $n$. It is worth mentioning that [LR23]'s Theorem 15 shaves roughly one logarithmic factor in the error bound from our Theorem 3. On the other hand, Theorem 3 actually requires a looser condition than mentioned above (see (20)), which can improve its guarantees in a wider range of parameters.

Finally, we briefly contextualize our population-level localization framework in regard to previous localization schemes proposed by [FKT20]. The two localization schemes in [FKT20] (see Sections 4.1 and 5.1 of that work) both follow the same strategy of gradually improving distance bounds to a minimizer in phases. However, their implementation is qualitatively different than our Algorithm 2, preventing their direct application in our algorithm. For instance, Section 4.1 of [FKT20] does not use strong convexity and, therefore cannot take advantage of generalization bounds afforded to

strongly convex losses (see discussion in [SSSS09]). On the other hand, the scheme in Section 5.1 of [FKT20] serves a different purpose than Algorithm 2, aiming to solve strongly convex optimization by reducing it to non-strongly convex optimization; our Algorithm 2, on the other hand, directly targets non-strongly convex optimization as its goal. We view our approach as complementary to these prior frameworks and are optimistic it will find further utility in applications.

## 2 Preliminaries

**General notation.** We use $[d]$ to denote the set $\{i \in \mathbb{N} \mid i \leq d\}$. We use $\text{sign}(x) \in \{\pm 1\}$ to denote the sign for $x \in \mathbb{R}$, with $\text{sign}(0) = 1$. We use $\mathcal{N}(\mu, \Sigma)$ to denote the multivariate normal distribution of specified mean and covariance. We denote the all-ones and all-zeroes vectors of dimension $d$ by $\mathbb{1}_d$ and $\mathbb{0}_d$. We use $\|\cdot\|$ to denote the Euclidean ($\ell_2$) norm. We use $\mathbf{I}_d$ to denote the identity matrix on $\mathbb{R}^d$. We use $\mathbb{B}(C)$ to denote the $\ell_2$ ball of radius $C$, and for $x \in \mathbb{R}^d$, $\mathbb{B}(x, C)$ is used to denote $\{x' \in \mathbb{R}^d \mid \|x' - x\| \leq C\}$. For a set $\mathcal{X} \subseteq \mathbb{R}^d$, we let $\text{diam}(\mathcal{X}) := \sup_{x, x' \in \mathcal{X}} \|x - x'\|$, and we let $\Pi_\mathcal{X}(x)$ denote the Euclidean projection of $x$ to $\mathcal{X}$, i.e. $\text{argmin}_{x' \in \mathcal{X}} \|x' - x\|$, which exists and is unique when $\mathcal{X}$ is compact. We use $f_\mathcal{X}$ to denote the restriction of a function $f$ to $\mathcal{X}$, i.e.

$$f_\mathcal{X}(x) = \begin{cases} f(x) & x \in \mathcal{X} \\ \infty & x \notin \mathcal{X} \end{cases}. \tag{4}$$

For $x \in \mathbb{R}^d$, we use $\Pi_C(x)$ as shorthand for $\Pi_{\mathbb{B}(C)}(x)$, i.e. $\Pi_C(X)$ denotes the clipped vector $x \cdot \min(\frac{C}{\|x\|}, 1)$. We say two datasets $\mathcal{D}, \mathcal{D}'$ are *neighboring* if they differ in one entry, and $|\mathcal{D}| = |\mathcal{D}'|$. We say $x \in \mathcal{X}$ is an $\varepsilon$-approximate minimizer to $f : \mathcal{X} \to \mathbb{R}$ if $f(x) - \inf_{x^\star \in \mathcal{X}} f(x^\star) \leq \varepsilon$. For two densities $\mu, \nu$ on the same probability space, and $\alpha > 1$, we define the $\alpha$-Rényi divergence

$$D_\alpha(\mu \| \nu) := \frac{1}{\alpha - 1} \log \left( \int \left( \frac{\mu(\omega)}{\nu(\omega)} \right)^\alpha \mathrm{d}\nu(\omega) \right).$$

For an event $\mathcal{E}$ on a probability space clear from context, we let $\mathbb{1}_\mathcal{E}$ denote the 0-1 indicator of $\mathcal{E}$. We say $f : \mathcal{X} \to \mathbb{R}$ is $L$-Lipschitz if $|f(x) - f(x')| \leq L \|x - x'\|$ for all $x, x' \in \mathcal{X}$; if $f$ is differentiable and convex, an equivalent characterization is $\|\nabla f(x)\| \leq L$ for all $x \in \mathcal{X}$. We say $f : \mathcal{X} \to \mathbb{R}$ is $\mu$-strongly convex if $f(\lambda x' + (1-\lambda)x) \leq \lambda f(x') + (1-\lambda)f(x) - \frac{\mu\lambda(1-\lambda)}{2} \|x - x'\|^2$ for all $x, x' \in \mathcal{X}$. We say differentiable $f : \mathcal{X} \to \mathbb{R}$ is $\beta$-smooth if for all $x, x' \in \mathcal{X}$, $\|\nabla f(x) - \nabla f(x')\| \leq \beta \|x - x'\|$.

**Differential privacy.** We begin with a definition of standard differential privacy.

**Definition 1** (Differential privacy). *Let $\varepsilon \geq 0$, $\delta \in [0, 1]$. We say a mechanism (randomized algorithm) $\mathcal{M} : \mathcal{S}^n \to \Omega$ satisfies $(\varepsilon, \delta)$-differential privacy (alternatively, $\mathcal{M}$ is $(\varepsilon, \delta)$-DP) if for any neighboring $\mathcal{D}, \mathcal{D}' \in \mathcal{S}^n$, and any $S \subseteq \Omega$, $\Pr[\mathcal{M}(\mathcal{D}) \in S] \leq \exp(\varepsilon)\Pr[\mathcal{M}(\mathcal{D}') \in S] + \delta$.*

*More generally, for random variables $X, Y \in \Omega$ satisfying $\Pr[X \in S] \leq \exp(\varepsilon)\Pr[Y \in S] + \delta$ for all $S \subseteq \Omega$, we say that $X, Y$ are $(\varepsilon, \delta)$-indistinguishable.*

Throughout the paper, other notions of differential privacy will frequently be useful for our accounting of privacy loss in our algorithms. For example, we define the following variants of DP.

**Definition 2** (Rényi DP). *Let $\alpha > 1$, $\varepsilon \geq 0$. We say a mechanism $\mathcal{M} : \mathcal{S}^n \to \Omega$ satisfies $(\alpha, \varepsilon)$-Rényi differential privacy (RDP) if for any neighboring $\mathcal{D}, \mathcal{D}' \in \mathcal{S}^n$, $D_\alpha(\mathcal{M}(\mathcal{D}) \| \mathcal{M}(\mathcal{D}')) \leq \varepsilon$.*

**Definition 3** (CDP). *Let $\rho \geq 0$. We say a mechanism $\mathcal{M} : \mathcal{S}^n \to \Omega$ satisfies $\rho$-concentrated differential privacy (alternatively, $\mathcal{M}$ satisfies $\rho$-CDP) if for any neighboring $\mathcal{D}, \mathcal{D}' \in \mathcal{S}^n$, and any $\alpha \geq 1$, $D_\alpha(\mathcal{M}(\mathcal{D}) \| \mathcal{M}(\mathcal{D}')) \leq \alpha\rho$.*

For an extended discussion of RDP and CDP and their properties, we refer the reader to [BS16, Mir17, BDRS18]. We summarize the main facts about these notions we use here.

**Lemma 1** ([Mir17]). *RDP has the following properties.*

1. *(Composition): Let $\mathcal{M}_1 : \mathcal{S}^n \to \Omega$ satisfy $(\alpha, \varepsilon_1)$-RDP and $\mathcal{M}_2 : \mathcal{S}^n \times \Omega \to \Omega'$ satisfy $(\alpha, \varepsilon_2)$-RDP for any input in $\Omega$. Then the composition of $\mathcal{M}_2$ and $\mathcal{M}_1$, i.e. the randomized algorithm which takes $\mathcal{D}$ to $\mathcal{M}_2(\mathcal{D}, \mathcal{M}_1(\mathcal{D}))$, satisfies $(\alpha, \varepsilon_1 + \varepsilon_2)$-RDP.*

2. *(RDP to DP): If $\mathcal{M}$ satisfies $(\alpha, \varepsilon)$-RDP, it satisfies $(\varepsilon + \frac{1}{\alpha-1}\log\frac{1}{\delta}, \delta)$-DP for all $\delta \in (0, 1)$.*

3. *(Gaussian mechanism): Let $f : \mathcal{S}^n \to \mathbb{R}^d$ be an $L$-sensitive randomized function for $L \geq 0$, i.e. for any neighboring $\mathcal{D}, \mathcal{D}'$, we have $\|f(\mathcal{D}) - f(\mathcal{D}')\| \leq L$. Then for any $\sigma > 0$, the mechanism which outputs $f(\mathcal{D}) + \xi$ for $\xi \sim \mathcal{N}(\mathbb{0}_d, \sigma^2 \mathbf{I}_d)$ satisfies $\frac{L^2}{2\sigma^2}$-CDP.*

**Private SCO.** Throughout the paper, we study the problem of private stochastic convex optimization (SCO) with heavy-tailed gradients. We first define the assumptions used in our algorithms.

**Assumption 1** ($k$-heavy-tailed distributions). *Let $\mathcal{X} \subset \mathbb{R}^d$ be a compact, convex set. Let $\mathcal{P}$ be a distribution over a sample space $\mathcal{S}$, such that each $s \in \mathcal{S}$ induces a continuously-differentiable, convex, $L_s$-Lipschitz loss function $f(\cdot; s) : \mathcal{X} \to \mathbb{R}$,[6] where $L_s := \max_{x \in \mathcal{X}} \|\nabla f(x; s)\|$ is unknown. For $k \in \mathbb{N}$ satisfying $k \geq 2$, we say $\mathcal{P}$ satisfies the $k$-heavy tailed assumption if, for a sequence of monotonically nondecreasing $\{G_j\}_{j \in [k]}$,[7] we have $\mathbb{E}_{s \sim \mathcal{P}}[L_s^j] \leq G_j^j < \infty$ for all $j \in [k]$.*

In Appendix B, we consider a variant of Assumption 1 where we have explicit access to upper bounds on the Lipschitz constants $L_s$, formalized in Assumption 2. Our goal is to approximately optimize a population loss over sample functions satisfying Assumptions 1 or 2, formalized in the following.

**Definition 4** ($k$-heavy-tailed private SCO). *In the $k$-heavy-tailed private SCO problem, $\mathcal{X} \subset \mathbb{R}^d$ is a compact, convex set with $\mathrm{diam}(\mathcal{X}) = D$. Further, $\mathcal{P}$ is a distribution over a sample space $\mathcal{S}$ satisfying Assumption 1. Our goal is to design an algorithm which provides an approximate minimizer in expectation to the population loss, $F_{\mathcal{P}}(x) := \mathbb{E}_{s \sim \mathcal{P}}[f(x; s)]$, subject to satisfying differential privacy. We say such an algorithm queries $N$ sample gradients if it queries $\nabla f(x; s)$ for $N$ different pairs $(x, s) \in \mathcal{X} \times \mathcal{S}$. If $\mathcal{P}$ further satisfies Assumption 2, we call the corresponding problem the* known Lipschitz $k$-heavy-tailed private SCO *problem.*

We first observe the following consequence of Assumption 1, deferring a proof to Appendix A.

**Lemma 2.** *Let $\mathcal{P}$ be a distribution over $\mathcal{S}$ satisfying Assumption 1. Then $F_{\mathcal{P}}$ is $G_1$-Lipschitz.*

We require the following claim which bounds the bias of clipped heavy-tailed distributions.

**Fact 1** ([BD14], Lemma 3). *Let $k > 1$ and $X \in \mathbb{R}^d$ be a random vector with $\mathbb{E}[\|X\|^k] \leq G^k$. Then,*

$$\mathbb{E} \|\Pi_C(X) - X\| \leq \mathbb{E}[\|X\| \, \mathbb{0}_{\|X\| \geq C}] \leq \frac{G^k}{(k-1)C^{k-1}}.$$

We also use the following standard claim on geometric aggregation.

**Fact 2** ([KLL+23], Claim 1). *Let $S := \{x_i\}_{i \in [k]} \subset \mathbb{R}^d$ have the property that for (unknown) $z \in \mathbb{R}^d$, $|\{i \in [k] \mid \|x_i - z\| \leq R\}| \geq 0.51k$ for some $R \geq 0$. There is an algorithm* Aggregate *which runs in time $O(dk^2)$ and outputs $x \in S$ such that $\|x - z\| \leq 3R$.*

Finally, given a dataset $\mathcal{D} \in \mathcal{S}^*$ of arbitrary size, and $\lambda \geq 0$, we use the following shorthand to denote the regularized empirical risk minimization (ERM) objective corresponding to the dataset:

$$F_{\mathcal{D}, \lambda}(x) := \frac{1}{|\mathcal{D}|} \sum_{s \in \mathcal{D}} f(x; s) + \frac{\lambda}{2} \|x\|^2. \tag{5}$$

When $\lambda = 0$, we simply denote the function above by $F_{\mathcal{D}}(x) := \frac{1}{|\mathcal{D}|} \sum_{s \in \mathcal{D}} f(x; s)$.

## 3 Heavy-Tailed Private SCO

In this section, we obtain near-optimal algorithms for the problem in Definition 4 using a new *population-level localization* framework, combined with geometric aggregation for boosting weak subproblem solvers to succeed with high probability (Fact 2). Our algorithm's main ingredient, in Section 3.1, is a clipped DP-SGD subroutine for privately minimizing a regularized ERM subproblem, under a condition on a randomly sampled dataset holding with constant probability. Next, in Section 3.2 we show that our algorithm from Section 3.1 returns points near the minimizer of a

---

[6]The assumed moment bounds shows that $f(\cdot; s)$ has a finite Lipschitz constant, except for a probability-zero set of $s$. Moreover, convex functions are differentiable almost everywhere. Therefore, if $f(\cdot; s)$ is Lipschitz, perturbing its first argument by an infinitesimal Gaussian makes it differentiable there with probability 1, and negligibly affects the function value. We thus assume for simplicity that $f(\cdot; s)$ is differentiable everywhere.

[7]This assumption is without loss of generality by Jensen's inequality.

regularized loss function over the population, using generalization arguments. Finally, we develop our population-level localization scheme in Section 3.3, and combine it with our subproblem solver to give our overall method for heavy-tailed private SCO. Several proofs and a generalization to strongly convex functions (Corollary 3) are deferred to Appendix A.

## 3.1 Strongly convex DP-ERM solver

We give a parameterized subroutine for minimizing a DP-ERM objective $F_{\mathcal{D},\lambda}(x)$ associated with a dataset $\mathcal{D}$ and a regularization parameter $\lambda \geq 0$ (recalling the definition (5)). In this section only, for notational convenience we identify elements of $\mathcal{D}$ with $[n]$ where $n := |\mathcal{D}|$, so we will also write

$$F_{\mathcal{D},\lambda}(x) := \frac{1}{n} \sum_{i \in [n]} f_i(x) + \frac{\lambda}{2} \|x\|^2,$$

i.e. we let $f_i(\cdot) := f(\cdot; s)$ where $s \in \mathcal{D}$ is the element identified with $i \in [n]$. Our subroutine is a clipped DP-SGD algorithm (Algorithm 1), which only clips the heavy-tailed portion of $\nabla F_{\mathcal{D},\lambda}$ (i.e. the sample gradients), and leaves both the regularization and additive noise unchanged. The utility of Algorithm 1 is parameterized by the following function of the dataset:

$$b_{\mathcal{D}} := \max_{x \in \mathcal{X}} \left\| \frac{1}{n} \sum_{i \in [n]} \nabla f_i(x) - \frac{1}{n} \sum_{i \in [n]} \Pi_C(\nabla f_i(x)) \right\|. \tag{6}$$

In other words, $b_{\mathcal{D}}$ denotes the maximum bias incurred by the clipped gradient of $F_{\mathcal{D}}$ when compared to the true gradient, over points in $\mathcal{X}$; note the maximum is achieved as $\mathcal{X}$ is compact.

We are now ready to state our algorithm, Clipped-DP-SGD, as Algorithm 1.

---

**Algorithm 1:** Clipped-DP-SGD$(\mathcal{D}, C, \lambda, \{\eta_t\}_{t \in [T]}, \sigma^2, T, r, \mathcal{X})$

**1 Input:** Dataset $\mathcal{D} \in \mathcal{S}^n$, clip threshold $C \in \mathbb{R}_{\geq 0}$, regularization $\lambda \in \mathbb{R}_{\geq 0}$, step sizes $\{\eta_t\}_{t \in [T]} \subset \mathbb{R}_{\geq 0}$, noise $\sigma^2 \in \mathbb{R}_{\geq 0}$, iteration count $T \in \mathbb{N}$, radius $r \in \mathbb{R}_{\geq 0}$, domain $\mathcal{X} \subset \mathbb{B}(r)$ with $\mathcal{X} \ni \mathbb{0}_d$

**2** $x_0 \leftarrow \mathbb{0}_d$

**3 for** $0 \leq t < T$ **do**

**4** $\quad \xi_t \sim \mathcal{N}(\mathbb{0}_d, \sigma^2 \mathbf{I}_d)$

**5** $\quad \hat{g}_t \leftarrow \frac{1}{n} \sum_{i \in [n]} \Pi_C(\nabla f_i(x_t))$

**6** $\quad x_{t+1} \leftarrow \operatorname{argmin}_{x \in \mathcal{X}_r} \{\eta_t \langle \hat{g}_t + \xi_t, x \rangle + \frac{\eta_t \lambda}{2} \|x\|^2 + \frac{1}{2} \|x - x_t\|^2\}$

**7 end**

**8 Return:** $\hat{x} \leftarrow \frac{\sum_{0 \leq t < T}(t+4)x_t}{\sum_{0 \leq t < T}(t+4)}$

---

We provide the following guarantee on Clipped-DP-SGD, by modifying an analysis of [LSB12].

**Proposition 1.** *Let* $\rho \geq 0$, *and* $\hat{x}$ *be the output of* Clipped-DP-SGD *with* $\eta_t \leftarrow \frac{4}{\lambda(t+1)}$ *for all* $0 \leq t < T$, $\sigma^2 \leftarrow \frac{2C^2 T}{n^2 \rho}$, *and* $T \geq \max(n, \frac{n^2 \rho}{d})$. Clipped-DP-SGD *satisfies* $\rho$-CDP, *and*

$$\mathbb{E}[F_{\mathcal{D},\lambda}(\hat{x}) - F_{\mathcal{D},\lambda}(x^\star)] \leq \frac{32C^2 d}{\lambda n^2 \rho} + \frac{b_{\mathcal{D}}^2}{\lambda} + \frac{7\lambda r^2}{n}, \text{ where } x^\star := \operatorname*{argmin}_{x \in \mathcal{X}} F_{\mathcal{D},\lambda}(x).$$

For ease of use of Proposition 1, we now provide a simple bound on $b_{\mathcal{D}}$ which holds with constant probability from a dataset drawn from a distribution satisfying Assumption 1.

**Lemma 3.** *Let* $\mathcal{D} \sim \mathcal{P}^n$, *where* $\mathcal{P}$ *is a distribution over* $\mathcal{S}$ *satisfying Assumption 1. With probability at least* $\frac{4}{5}$, *denoting* $b_{\mathcal{D}}$ *as in* (26), *we have*

$$b_{\mathcal{D}} \leq \frac{5G_k^k}{(k-1)C^{k-1}}.$$

We therefore have the following corollary of Proposition 1 and Lemma 3.

**Corollary 1.** *Let $\mathcal{D} \sim \mathcal{P}^n$, where $\mathcal{P}$ is a distribution over $\mathcal{S}$ satisfying Assumption 1, and let $x_{\mathcal{D},\lambda}^{\star} :=$ $\arg\min_{x \in \mathcal{X}} F_{\mathcal{D},\lambda}(x)$, following (5). If we run Clipped-DP-SGD with parameters in Proposition 1 and $C \leftarrow G_k \cdot (\frac{25n^2\rho}{32d})^{\frac{1}{2k}}$, Clipped-DP-SGD is $\rho$-CDP, and there is a universal constant $C_{\mathrm{erm}}$ such that with probability $\geq \frac{3}{5}$ over the randomness of $\mathcal{D}$ and Clipped-DP-SGD, $\hat{x}$, the output of Clipped-DP-SGD, satisfies*

$$\left\| \hat{x} - x_{\mathcal{D},\lambda}^{\star} \right\| \leq C_{\mathrm{erm}} \left( \frac{G_k}{\lambda} \left( \frac{\sqrt{d}}{n\sqrt{\rho}} \right)^{1 - \frac{1}{k}} + \frac{r}{\sqrt{n}} \right).$$

Clipped-DP-SGD *queries at most* $\max(n^2, \frac{n^3\rho}{d})$ *sample gradients (using samples in $\mathcal{D}$).*

*Proof.* Condition on the conclusion of Lemma 3, which holds with probability $\frac{4}{5}$. Therefore, Markov's inequality shows that with probability at least $\frac{3}{5}$, after a union bound with Proposition 1,

$$\frac{\lambda}{2} \left\| \hat{x} - x_{\mathcal{D},\lambda}^{\star} \right\|^2 \leq F_{\mathcal{D},\lambda}(\hat{x}) - F_{\mathcal{D},\lambda}(x_{\mathcal{D},\lambda}^{\star})$$

$$\leq \frac{160C^2 d}{\lambda n^2 \rho} + \frac{125G_k^{2k}}{\lambda C^{2(k-1)}} + \frac{7\lambda r^2}{n} \leq \frac{320G_k^2}{\lambda} \left( \frac{d}{n^2\rho} \right)^{1 - \frac{1}{k}} + \frac{7\lambda r^2}{n},$$

where we used strong convexity in the first inequality, and plugged in our choice of $C$ in the last. The conclusion follows by rearranging the above display, and using $\sqrt{a^2 + b^2} \leq a + b$ for $a, b \in \mathbb{R}_{\geq 0}$. $\square$

## 3.2 Localizing regularized population loss minimizers

Here, we use generalization arguments from the SCO literature to show how that our algorithm Clipped-DP-SGD from Section 3.1 acts as an oracle which, with constant probability, returns a point near the minimizer of a regularized population loss. We begin with a standard helper statement.

**Lemma 4.** *Let $\lambda \geq 0$, let $\mathcal{P}$ be a distribution over $\mathcal{S}$ satisfying Assumption 1, let $\bar{x} \in \mathcal{X}$ where $\mathcal{X} \subset \mathbb{R}^d$ is compact and convex, and let*

$$x_{\lambda,\bar{x}}^{\star} := \arg\min_{x \in \mathcal{X}} \left\{ F_{\mathcal{P}}(x) + \frac{\lambda}{2} \|x - \bar{x}\|^2 \right\}, \quad \text{where } F_{\mathcal{P}}(x) := \mathbb{E}_{s \sim \mathcal{P}} \left[ f(x; s) \right]. \tag{7}$$

*Then $\|\bar{x} - x_{\lambda,\bar{x}}^{\star}\| \leq \frac{2G_1}{\lambda}$.*

Next, we apply a result on generalization due to [LR23] to bound the expected distance between a restricted empirical regularized minimizer and the minimizer of the population variant in (7).

**Lemma 5.** *Let $\lambda \geq 0$, let $\mathcal{D} \sim \mathcal{P}^n$ where $\mathcal{P}$ is a distribution over $\mathcal{S}$ satisfying Assumption 1, and let $\bar{x} \in \mathcal{X}$ where $\mathcal{X} \subset \mathbb{R}^d$ is compact and convex. Following notation (4), (5), let*

$$y := \arg\min_{x \in \mathcal{X}} \left\{ [F_{\mathcal{D}}]_{\mathbb{B}(\bar{x},r)} (x) + \frac{\lambda}{2} \|x - \bar{x}\|^2 \right\}, \quad \text{for } r := \frac{2G_1}{\lambda}$$

*and let $x_{\lambda,\bar{x}}^{\star}$ be defined as in (7). Then with probability $\geq 0.95$ over the randomness of $\mathcal{D} \sim \mathcal{P}^n$,*

$$\left\| y - x_{\lambda,\bar{x}}^{\star} \right\|_2 \leq \frac{90G_2}{\lambda\sqrt{n}}.$$

**Corollary 2.** *Let $\mathcal{D} \sim \mathcal{P}^n$, where $\mathcal{P}$ is a distribution over $\mathcal{S}$ satisfying Assumption 1, and let $\bar{x} \in \mathcal{X}$ where $\mathcal{X} \subset \mathbb{R}^d$ is compact and convex. Let $\lambda \geq 0$ and define $x_{\lambda,\bar{x}}^{\star}$ as in (7). There is a $\rho$-CDP algorithm $\mathcal{A}$ which queries $\max(n^2, \frac{n^3\rho}{d})$ sample gradients (using samples in $\mathcal{D}$). With probability 0.55 over the randomness of $\mathcal{A}$ and $\mathcal{D}$, $\mathcal{A}$ returns $\hat{x}$ satisfying, for a universal constant $C_{\mathrm{reg\text{-}pop}}$,*

$$\left\| \hat{x} - x_{\lambda,\bar{x}}^{\star} \right\| \leq C_{\mathrm{reg\text{-}pop}} \left( \frac{G_k}{\lambda} \left( \frac{\sqrt{d}}{n\sqrt{\rho}} \right)^{1 - \frac{1}{k}} + \frac{G_2}{\lambda\sqrt{n}} \right).$$

*Proof.* Condition on the conclusion of Lemma 5 holding for our dataset, which loses $0.05$ in the failure probability. Next, consider the guarantee of Corollary 1, when applied to the truncated and shifted functions, $\tilde{f}(x;s) \leftarrow f_{\mathbb{B}(\bar{x},r)}(x - \bar{x};s)$, where $r$ is set as in Lemma 5. It shows that with probability $\frac{3}{5}$, $\|\hat{x} + \bar{x} - y\| = O(\frac{G_k}{\lambda}(\frac{\sqrt{d}}{n\sqrt{\rho}})^{1-\frac{1}{k}} + \frac{\sqrt{\lambda}r}{\sqrt{n}})$, for the point $\hat{x}$ returned by the algorithm, and $y$ the exact minimizer of the empirical loss restricted to $\mathbb{B}(\bar{x},r)$. Therefore, the conclusion follows by overloading $\hat{x} \leftarrow \hat{x} + \bar{x}$, applying the triangle inequality with the conclusions of Corollary 1 and 2, and taking a union bound over their failure probabilities. $\qquad\square$

### 3.3 Population-level localization

In this section, we provide a generic population-level localization scheme for stochastic convex optimization, which may be of broader interest. Our localization scheme is largely patterned off of the analogous localization methods developed by [FKT20], but directly argues about contraction to population-level regularized minimizers (as opposed to empirical minimizers), which makes it compatible with our framework in Section 3.1 and 3.2, specifically the guarantees of Corollary 2.

---

**Algorithm 2:** Population-Localize$(x_0, \mathcal{P}, \lambda, I)$

1 **Input:** Initial point $x_0 \in \mathcal{X}$, distribution $\mathcal{P}$ over samples in $\mathcal{S}$, for $\mathcal{X}, \mathcal{S}$ inducing a $k$-heavy-tailed DP-SCO problem as in Definition 4, with a population loss $F_{\mathcal{P}} := \mathbb{E}_{s \sim \mathcal{S}}[f(\cdot;s)], \lambda \geq 0, I \in \mathbb{N}$

2 **for** $i \in [I]$ **do**

3 $\quad \lambda_i \leftarrow \lambda \cdot 32^i$

4 $\quad x_i \leftarrow$ any point satisfying

$$\|x_i - x_i^\star\| \leq \frac{\Delta 4^i}{\lambda_i}, \text{ where } x_i^\star := \operatorname*{argmin}_{x \in \mathcal{X}} \left\{ F_{\mathcal{P}}(x) + \frac{\lambda_i}{2} \|x - x_{i-1}\|^2 \right\} \qquad (8)$$

5 **end**

6 **Return:** $x_I$

---

We briefly discuss the role of the hyperparameters $\lambda, \Delta$ in Algorithm 2 for clarity. The parameter $\Delta$ scales with the error guarantee of our regularized ERM solver; in particular, it will be determined by the bound in Corollary 2. The parameter $\lambda$ specifies an initial regularization amount that will later be tuned to trade off the terms in the following Proposition 2.

**Proposition 2.** *Following notation of Algorithm 2, let $x^\star := \operatorname{argmin}_{x \in \mathcal{X}} F_{\mathcal{P}}(x)$. Then,*

$$F_{\mathcal{P}}(x_I) - F_{\mathcal{P}}(x^\star) \leq \frac{G_1 \Delta}{\lambda 8^I} + \frac{\Delta^2}{4\lambda} + \frac{\lambda D^2}{2}.$$

*In particular, choosing $\lambda$ to optimize this bound, we have*

$$F_{\mathcal{P}}(x_I) - F_{\mathcal{P}}(x^\star) \leq 2D\sqrt{\frac{G_1 \Delta}{8^I}} + D\Delta.$$

*Proof.* We denote $x_0^\star := x^\star$ throughout the proof. First, we expand

$$F_{\mathcal{P}}(x_I) - F_{\mathcal{P}}(x_0^\star) = F_{\mathcal{P}}(x_I) - F_{\mathcal{P}}(x_I^\star) + F_{\mathcal{P}}(x_I^\star) - F_{\mathcal{P}}(x_0^\star)$$
$$= F_{\mathcal{P}}(x_I) - F_{\mathcal{P}}(x_I^\star) + \sum_{i \in [I]} F_{\mathcal{P}}(x_i^\star) - F_{\mathcal{P}}(x_{i-1}^\star).$$

Moreover, for each $i \in [I]$, since $x_i^\star$ minimizes $F_{\mathcal{P}}(x) + \frac{\lambda_i}{2}\|x - x_{i-1}\|^2$,

$$F_{\mathcal{P}}(x_i^\star) \leq F_{\mathcal{P}}(x_i^\star) + \frac{\lambda_i}{2} \|x_i^\star - x_{i-1}\|^2 \leq F_{\mathcal{P}}(x_{i-1}^\star) + \frac{\lambda_i}{2} \|x_{i-1}^\star - x_{i-1}\|^2.$$

Combining the above two displays, and using that $F_{\mathcal{P}}$ is $G_1$-Lipschitz (Lemma 2), we have

$$F_{\mathcal{P}}(x_I) - F_{\mathcal{P}}(x^\star) \le G_1 \left\| x_I - x_I^\star \right\| + \sum_{i \in [I]} \frac{\lambda_i}{2} \left\| x_{i-1}^\star - x_{i-1} \right\|^2$$

$$\le \frac{G_1 \Delta}{\lambda 8^I} + \sum_{i \in [I-1]} \frac{\Delta^2 16^i}{2\lambda_i} + \frac{\lambda D^2}{2} \le \frac{G_1 \Delta}{\lambda 8^I} + \frac{\Delta^2}{4\lambda} + \frac{\lambda D^2}{2},$$

where we used the diameter bound assumption $\mathsf{diam}(\mathcal{X}) = D$, as in Definition 4. $\qquad\square$

In particular, note that Corollary 2 shows that by using $n$ samples from $\mathcal{P}$ and a CDP budget of $\rho$, with constant probability, we can satisfy the requirement (8) with $\Delta 4^i = O(G_k(\frac{\sqrt{d}}{n\sqrt{\rho}})^{1-\frac{1}{k}} + \frac{G_2}{\sqrt{n}})$. By plugging this guarantee into the aggregation subroutine in Fact 2, we have our SCO algorithm.

---

**Algorithm 3:** Aggregate-ERM$(\bar{x}, \lambda, J, \rho, \{s_\ell\}_{\ell \in [nJ]}, R)$

---

1 **Input:** Regularization center $\bar{x} \in \mathcal{X}$, regularization $\lambda \in \mathbb{R}_{\ge 0}$, sample split parameter $J \in \mathbb{N}$, privacy parameter $\rho \in \mathbb{R}_{\ge 0}$, samples $\{s_\ell\}_{\ell \in [nJ]} \subset \mathcal{S}$, distance bound $R \in \mathbb{R}_{\ge 0}$

2 **for** $j \in [J]$ **do**

3 $\quad \mathcal{D}^j \leftarrow \{s_\ell\}_{(j-1)n < \ell \le jn}$ for all $j \in [J]$

4 $\quad x^j \leftarrow$ result of Corollary 2 using $\mathcal{D}^j$, on loss defined by $\bar{x}, \lambda$ with privacy parameter $\rho$, i.e., $\quad x^j$ is a point satisfying, with probability 0.55, for a universal constant $C_{\text{reg-pop}}$,

$$\left\| x^j - x_{\lambda,\bar{x}}^\star \right\| \le C_{\text{reg-pop}} \left( \frac{G_k}{\lambda} \left( \frac{\sqrt{d}}{n\sqrt{\rho}} \right)^{1-\frac{1}{k}} + \frac{G_2}{\lambda\sqrt{n}} \right)$$

5 **end**

6 $x \leftarrow \mathsf{Aggregate}(\{x^j\}_{j \in [J]}, R)$ (see Fact 2)

7 **Return:** $x$

---

**Theorem 1.** *Consider an instance of $k$-heavy-tailed private SCO, following notation in Definition 4, let $x^\star := \arg\min_{x \in \mathcal{X}} F_{\mathcal{P}}(x)$, and let $\rho \ge 0$, $\delta \in (0, 1)$. Algorithm 2 using Algorithm 3 in Line 5 is a $\rho$-CDP algorithm which draws $\mathcal{D} \sim \mathcal{P}^n$, queries $C_{\text{sco}} \max(n^2, \frac{n^3 \rho}{d})$ sample gradients (using samples in $\mathcal{D}$) for a universal constant $C_{\text{sco}}$, and outputs $x \in \mathcal{X}$ satisfying, with probability $\ge 1 - \delta$,*

$$F_{\mathcal{P}}(x) - F_{\mathcal{P}}(x^\star) \le C_{\text{sco}} \left( G_k D \cdot \left( \frac{\sqrt{d} \log\left(\frac{1}{\delta}\right)}{n\sqrt{\rho}} \right)^{1-\frac{1}{k}} + G_2 D \cdot \sqrt{\frac{\log\left(\frac{1}{\delta}\right)}{n}} \right).$$

## 4 Conclusion

In this work, we consider the DP-SCO with heavy-tailed gradients. When the $k$-th moments of gradients are bounded, we propose the population-level localization framework and attain near-optimal excess loss $G_2 D \cdot \sqrt{\frac{\log(\frac{1}{\delta})}{n}} + G_k D \cdot (\frac{\sqrt{d} \log(\frac{1}{\delta})}{n\sqrt{\rho}})^{1-\frac{1}{k}}$ with probability at least $1 - \delta$ and satisfy $\rho$-CDP. We can achieve a tight rate for high-probability SCO under a bounded-variance gradient estimator parameterization by applying the population-level localization framework. Moreover, we improve this basic result under additional assumptions, including an optimal algorithm under a known-Lipschitz constant assumption, a near-linear time algorithm for smooth functions, and an optimal linear time algorithm for smooth generalized linear models, with interesting techniques adapted to each setting.

It leaves many intriguing open problems in this direction. For example, can we design near-linear time algorithms for non-smooth functions? Can the population-level localization framework be applied to solve other problems? Can we establish a high-probability lower bound or eliminate the additional logarithmic term if we are only concerned with the excess bound in expectation? Can we evaluate the algorithm's performance through numerical simulations or real-world datasets? We leave these questions for future research.

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

# A Deferred proofs from the main body

## A.1 Deferred proofs from Section 2

**Lemma 2.** *Let $\mathcal{P}$ be a distribution over $\mathcal{S}$ satisfying Assumption 1. Then $F_{\mathcal{P}}$ is $G_1$-Lipschitz.*

*Proof.* This follows from the derivation

$$\max_{x \in \mathcal{X}} \|\mathbb{E}_{s \sim \mathcal{P}} \left[ \nabla f(x; s) \right]\| \leq \max_{x \in \mathcal{X}} \mathbb{E}_{s \sim \mathcal{P}} \|\nabla f(x; s)\| \leq \mathbb{E}_{s \sim \mathcal{P}} \max_{x \in \mathcal{X}} \|\nabla f(x; s)\| \leq G_1.$$

$\square$

## A.2 Deferred proofs from Section 3

**Proposition 1.** *Let $\rho \geq 0$, and $\hat{x}$ be the output of* Clipped-DP-SGD *with $\eta_t \leftarrow \frac{4}{\lambda(t+1)}$ for all $0 \leq t < T$, $\sigma^2 \leftarrow \frac{2C^2 T}{n^2 \rho}$, and $T \geq \max(n, \frac{n^2 \rho}{d})$.* Clipped-DP-SGD *satisfies $\rho$-CDP, and*

$$\mathbb{E}[F_{\mathcal{D},\lambda}(\hat{x}) - F_{\mathcal{D},\lambda}(x^\star)] \leq \frac{32 C^2 d}{\lambda n^2 \rho} + \frac{b_{\mathcal{D}}^2}{\lambda} + \frac{7 \lambda r^2}{n}, \text{ where } x^\star := \underset{x \in \mathcal{X}}{\operatorname{argmin}} F_{\mathcal{D},\lambda}(x).$$

*Proof.* For the privacy claim, note that each call to Line 3 is a postprocessing of a $\frac{2C}{n}$-sensitive statistic of the dataset $\mathcal{D}$, since neighboring databases can only change $\frac{1}{n} \sum_{i \in [n]} \Pi_C(\nabla f_i(x_t))$ by $\frac{2C}{n}$ in the $\ell_2$ norm, via the triangle inequality. Therefore, applying the first and third parts of Lemma 1 shows that after $T$ iterations, the CDP of the mechanism is at most $T \cdot \frac{2C^2}{n^2 \sigma^2} \leq \rho$.

We next prove the utility claim. For each $0 \leq t \leq T$, denote

$$\Delta_t := \mathbb{E}\left[ F_{\mathcal{D},\lambda}(x_t) - F_{\mathcal{D},\lambda}(x^\star) \right], \ \Phi_t := \mathbb{E}\left[ \frac{1}{2} \|x_t - x^\star\|^2 \right], \ g_t := \nabla F_{\mathcal{D}}(x_t),$$

where all expectations are only over randomness used by the algorithm, and not the randomness in sampling $\mathcal{D}$. First-order optimality applied to the definition of $x_{t+1}$ implies, for all $0 \leq t < T$,

$$\langle \hat{g}_t + \xi_t, x_t - x^\star \rangle + \langle \lambda x_{t+1}, x_{t+1} - x^\star \rangle \leq \frac{1}{2\eta_t} \left( \|x_t - x^\star\|^2 - \|x_{t+1} - x^\star\|^2 \right) + \frac{\eta_t}{2} \|\hat{g}_t + \xi_t\|^2.$$

Adding $\langle g_t - \hat{g}_t - \xi_t, x_t - x^\star \rangle$ to both sides and rearranging shows

$$
\begin{aligned}
&F_{\mathcal{D}}(x_t) + \frac{\lambda}{2} \|x_{t+1}\|^2 - F_{\mathcal{D},\lambda}(x^\star) + \frac{\lambda}{2} \|x_{t+1} - x^\star\|^2 \\
&\leq \langle g_t, x_t - x^\star \rangle + \langle \lambda x_{t+1}, x_{t+1} - x^\star \rangle \\
&\leq \frac{1}{2\eta_t} \left( \|x_t - x^\star\|^2 - \|x_{t+1} - x^\star\|^2 \right) + \frac{\eta_t}{2} \|\hat{g}_t + \xi_t\|^2 + \langle g_t - \hat{g}_t - \xi_t, x_t - x^\star \rangle \\
&\leq \frac{1}{2\eta_t} \left( \|x_t - x^\star\|^2 - \|x_{t+1} - x^\star\|^2 \right) + \eta_t C^2 + \eta_t \|\xi_t\|^2 + b_{\mathcal{D}} \|x_t - x^\star\| - \langle \xi_t, x_t - x^\star \rangle.
\end{aligned}
$$

In the first line, we used strong convexity of the function $\frac{\lambda}{2} \|x\|^2$, and in the last line, we used $\|a + b\|^2 \leq 2 \|a\|^2 + 2 \|b\|^2$ and the definitions of $C$ and $b_{\mathcal{D}}$. Next, adding $\frac{\lambda}{2}(\|x_t\|^2 - \|x_{t+1}\|^2)$ to both sides above and taking expectations over the first $t$ iterations yields

$$\Delta_t + \lambda \Phi_{t+1} \leq \frac{1}{\eta_t} \left( \Phi_t - \Phi_{t+1} \right) + \eta_t (C^2 + \sigma^2 d) + \frac{b_{\mathcal{D}}^2}{\lambda} + \frac{\lambda}{2} \Phi_t + \frac{\lambda}{2} \left( \mathbb{E} \|x_t\|^2 - \mathbb{E} \|x_{t+1}\|^2 \right),$$

where we used the Fenchel-Young inequality to bound $b_{\mathcal{D}} \|x_t - x^\star\| \leq \frac{b_{\mathcal{D}}^2}{\lambda} + \frac{\lambda}{4} \|x_t - x^\star\|^2$. Now, plugging in our step size schedule $\eta_t = \frac{4}{\lambda(t+1)}$, multiplying by $t + 4$, and rearranging shows

$$
\begin{aligned}
(t+4)\Delta_t \leq\ & \frac{\lambda(t+3)(t+4)}{4} \Phi_t - \frac{\lambda(t+5)(t+4)}{4} \Phi_{t+1} \\
& + \frac{4(t+4)}{\lambda(t+1)} \left( \frac{3C^2 T d}{n^2 \rho} \right) + \frac{(t+4) b_{\mathcal{D}}^2}{\lambda} + \frac{\lambda(t+4)}{2} \left( \mathbb{E} \|x_t\|^2 - \mathbb{E} \|x_{t+1}\|^2 \right),
\end{aligned}
$$

where we plugged in the choice of $\sigma^2$ and $T \geq \frac{n^2\rho}{d}$, so $C^2 \leq \frac{\sigma^2 d}{2}$. Summing the above for $0 \leq t < T$, using that all iterates and $x^\star$ lie in $\mathbb{B}(r)$, and dividing by $Z := \sum_{0 \leq t < T}(t+4) \geq \frac{T^2}{2}$, shows

$$\frac{1}{Z}\sum_{0 \leq t < T}(t+4)\Delta_t \leq \frac{3\lambda\Phi_0}{Z} + \frac{16C^2T^2d}{\lambda Z n^2 \rho} + \frac{b_{\mathcal{D}}^2}{\lambda} + \frac{\lambda}{2Z}\sum_{t \in [T]}\mathbb{E}\|x_t\|^2$$

$$\leq \frac{6\lambda r^2}{T^2} + \frac{32C^2 d}{\lambda n^2 \rho} + \frac{b_{\mathcal{D}}^2}{\lambda} + \frac{\lambda r^2}{T} \leq \frac{32C^2 d}{\lambda n^2 \rho} + \frac{b_{\mathcal{D}}^2}{\lambda} + \frac{7\lambda r^2}{T}.$$

The conclusion follows from convexity of $F_{\mathcal{D},\lambda}$, the definition of $\hat{x}$, and $T \geq n$. $\qquad\square$

**Lemma 3.** *Let $\mathcal{D} \sim \mathcal{P}^n$, where $\mathcal{P}$ is a distribution over $\mathcal{S}$ satisfying Assumption 1. With probability at least $\frac{4}{5}$, denoting $b_{\mathcal{D}}$ as in (26), we have*

$$b_{\mathcal{D}} \leq \frac{5G_k^k}{(k-1)C^{k-1}}.$$

*Proof.* For every $s \in \mathcal{S}$ let $x^\star(s) := \mathrm{argmax}_{x \in \mathcal{X}}\|\nabla f(x;s) - \Pi_C(\nabla f(x;s))\|_2$. Then, we have

$$\mathbb{E}_{\mathcal{D} \sim \mathcal{P}^n}[b_{\mathcal{D}}] = \mathbb{E}_{\mathcal{D} \sim \mathcal{P}^n}\left[\max_{x \in \mathcal{X}}\left\|\frac{1}{n}\sum_{i \in [n]}\nabla f_i(x) - \frac{1}{n}\sum_{i \in [n]}\Pi_C(\nabla f_i(x))\right\|\right]$$

$$\leq \frac{1}{n}\sum_{i \in [n]}\mathbb{E}_{\mathcal{D} \sim \mathcal{P}^n}\left[\max_{x \in \mathcal{X}}\|\nabla f_i(x) - \Pi_C(\nabla f_i(x))\|\right]$$

$$= \mathbb{E}_{s \sim \mathcal{P}}\left[\|\nabla f(x^\star(s);s) - \Pi_C(\nabla f(x^\star(s);s))\|\right] \leq \frac{\mathbb{E}\left[\|\nabla f(x^\star(s);s)\|^k\right]}{(k-1)C^{k-1}}$$

$$\leq \frac{G_k^k}{(k-1)C^{k-1}}.$$

The last line used independence of samples, used Fact 1 on the random vector $\nabla f(x^\star(s);s)$, and applied Assumption 1 with the definition of $x^\star(s)$. The conclusion uses Markov's inequality. $\quad\square$

**Lemma 4.** *Let $\lambda \geq 0$, let $\mathcal{P}$ be a distribution over $\mathcal{S}$ satisfying Assumption 1, let $\bar{x} \in \mathcal{X}$ where $\mathcal{X} \subset \mathbb{R}^d$ is compact and convex, and let*

$$x_{\lambda,\bar{x}}^\star := \mathrm{argmin}_{x \in \mathcal{X}}\left\{F_{\mathcal{P}}(x) + \frac{\lambda}{2}\|x - \bar{x}\|^2\right\}, \quad \text{where } F_{\mathcal{P}}(x) := \mathbb{E}_{s \sim \mathcal{P}}\left[f(x;s)\right]. \qquad (7)$$

*Then $\|\bar{x} - x_{\lambda,\bar{x}}^\star\| \leq \frac{2G_1}{\lambda}$.*

*Proof.* Let $r := \|\bar{x} - x^\star\|$. By strong convexity and the definition of $x_{\lambda,\bar{x}}^\star$,

$$\frac{\lambda r^2}{2} \leq F_{\mathcal{P}}(\bar{x}) - F_{\mathcal{P}}(x^\star) - \frac{\lambda}{2}\|x^\star - \bar{x}\|^2 \leq F_{\mathcal{P}}(\bar{x}) - F_{\mathcal{P}}(x^\star) \leq G_1 r.$$

Here, we used that $F_{\mathcal{P}}$ is $G_1$-Lipschitz (Lemma 2), and rearranging yields the conclusion. $\quad\square$

**Lemma 5.** *Let $\lambda \geq 0$, let $\mathcal{D} \sim \mathcal{P}^n$ where $\mathcal{P}$ is a distribution over $\mathcal{S}$ satisfying Assumption 1, and let $\bar{x} \in \mathcal{X}$ where $\mathcal{X} \subset \mathbb{R}^d$ is compact and convex. Following notation (4), (5), let*

$$y := \mathrm{argmin}_{x \in \mathcal{X}}\left\{[F_{\mathcal{D}}]_{\mathbb{B}(\bar{x},r)}(x) + \frac{\lambda}{2}\|x - \bar{x}\|^2\right\}, \quad \text{for } r := \frac{2G_1}{\lambda}$$

*and let $x_{\lambda,\bar{x}}^\star$ be defined as in (7). Then with probability $\geq 0.95$ over the randomness of $\mathcal{D} \sim \mathcal{P}^n$,*

$$\left\|y - x_{\lambda,\bar{x}}^\star\right\|_2 \leq \frac{90G_2}{\lambda\sqrt{n}}.$$

*Proof.* For each $f(x; s)$, define a restricted variant $\tilde{f}(x; s) := f_{\mathbb{B}(\bar{x},r)}(x; s)$, and let $\widetilde{F}_{\mathcal{P}} := \mathbb{E}_{s \sim \mathcal{S}} \tilde{f}(\cdot; s)$. Similarly, define $\widetilde{F}_{\mathcal{D}}$ to be the restricted variant of the empirical loss $F_{\mathcal{D}}$. Because $\widetilde{F}_{\mathcal{P}}$ is pointwise larger than $F_{\mathcal{P}}$ and $x^{\star}_{\lambda,\bar{x}} \in \mathbb{B}(\bar{x}, r)$ by Lemma 4, it is clear that

$$x^{\star}_{\lambda,\bar{x}} = \operatorname*{argmin}_{x \in \mathcal{X}} \left\{ \widetilde{F}_{\mathcal{P}}(x) + \frac{\lambda}{2} \|x - \bar{x}\|^2 \right\},$$

and $y$ is the minimizer of the empirical (restricted) variant of the above display. Moreover, each of the regularized functions $\tilde{f}(x; s) + \frac{\lambda}{2} \|x - \bar{x}\|^2$ has a Lipschitz constant at most $\lambda r = 2G_1$ larger than its unregularized counterpart in $\mathcal{X} \cap \mathbb{B}(\bar{x}, r)$, so these functions satisfy the moment bound in Assumption 1 for $j = 2$ with a bound of $2G_2^2 + 8G_1^2$. Now, applying Proposition 29, [LR23] yields

$$\mathbb{E}\left[ \left( \widetilde{F}_{\mathcal{P}}(y) + \frac{\lambda}{2} \|y - \bar{x}\|^2 \right) - \left( \widetilde{F}_{\mathcal{P}}(x^{\star}_{\lambda,\bar{x}}) + \frac{\lambda}{2} \|x^{\star}_{\lambda,\bar{x}} - \bar{x}\|^2 \right) \right]$$

$$= \mathbb{E}\left[ \left( \widetilde{F}_{\mathcal{D}}(y) + \frac{\lambda}{2} \|y - \bar{x}\|^2 \right) - \left( \widetilde{F}_{\mathcal{D}}(x^{\star}_{\lambda,\bar{x}}) + \frac{\lambda}{2} \|x^{\star}_{\lambda,\bar{x}} - \bar{x}\|^2 \right) \right]$$

$$+ \mathbb{E}\left[ \left( \widetilde{F}_{\mathcal{P}}(y) + \frac{\lambda}{2} \|y - \bar{x}\|^2 \right) - \left( \widetilde{F}_{\mathcal{D}}(y) + \frac{\lambda}{2} \|y - \bar{x}\|^2 \right) \right]$$

$$\leq 0 + \frac{4G_2^2 + 16G_1^2}{\lambda n} = \frac{4G_2^2 + 16G_1^2}{\lambda n} \leq \frac{20G_2^2}{\lambda n}.$$

The first equality used that $x^{\star}_{\lambda,\bar{x}}$ is independent of sampling $\mathcal{D}$, and the second used $\hat{x}$ is the empirical risk minimizer. The conclusion follows from Markov's inequality and strong convexity. $\square$

**Theorem 1.** *Consider an instance of $k$-heavy-tailed private SCO, following notation in Definition 4, let $x^{\star} := \operatorname{argmin}_{x \in \mathcal{X}} F_{\mathcal{P}}(x)$, and let $\rho \geq 0$, $\delta \in (0, 1)$. Algorithm 2 using Algorithm 3 in Line 5 is a $\rho$-CDP algorithm which draws $\mathcal{D} \sim \mathcal{P}^n$, queries $C_{\mathrm{sco}} \max(n^2, \frac{n^3 \rho}{d})$ sample gradients (using samples in $\mathcal{D}$) for a universal constant $C_{\mathrm{sco}}$, and outputs $x \in \mathcal{X}$ satisfying, with probability $\geq 1 - \delta$,*

$$F_{\mathcal{P}}(x) - F_{\mathcal{P}}(x^{\star}) \leq C_{\mathrm{sco}} \left( G_k D \cdot \left( \frac{\sqrt{d} \log\left(\frac{1}{\delta}\right)}{n \sqrt{\rho}} \right)^{1 - \frac{1}{k}} + G_2 D \cdot \sqrt{\frac{\log\left(\frac{1}{\delta}\right)}{n}} \right).$$

*Proof.* Throughout, we assume that $\frac{1}{\delta}$ is at least a large enough constant (where lossiness can be absorbed into $C_{\mathrm{sco}}$), and that $n$ is at least a sufficiently large constant multiple of $\log \frac{1}{\delta}$ (because the entire range of $F_{\mathcal{P}}$ is $\leq G_2 D$). We first handle the case where $\frac{1}{\delta}$ is larger than polylog$(n)$, deferring the case of small $\frac{1}{\delta}$ to the end of the proof. Let $I, J \in \mathbb{N}$ be chosen such that

$$I := \left\lfloor \log_2\left(\frac{n}{J}\right) \right\rfloor, \quad J \in \left[ 400 \log\left(\frac{I}{\delta}\right), 500 \log\left(\frac{I}{\delta}\right) \right],$$

which is achievable with $I = O(\log n)$ and $J = O(\log \frac{\log n}{\delta}) = O(\log \frac{1}{\delta})$. Let $m := \frac{n}{J}$, and assume without loss that $m$ is a power of 2, which we can guarantee by discarding $\leq \frac{1}{2}$ our samples, losing a constant factor in the claim. For each $i \in [I]$, let $m_i := \frac{m}{2^i}$. We subdivide $\mathcal{D}$ into $J$ portions, each with $m$ samples, and subdivide each portion into $I$ parts each with $m_i$ samples. For $j \in [J]$ and $i \in [I]$, we denote the samples corresponding to the $i^{\mathrm{th}}$ part of the $j^{\mathrm{th}}$ portion by $\mathcal{D}_i^j$, so

$$\bigcup_{i \in [I]} \bigcup_{j \in [J]} \mathcal{D}_i^j \subseteq \mathcal{D}, \ |\mathcal{D}_i^j| = m_i \text{ for all } j \in [J], \ \mathcal{D}_i^j \cap \mathcal{D}_{i'}^{j'} = \emptyset \text{ for all } (i, j) \neq (i', j').$$

Next, we show how to implement Line 5 in Algorithm 2, for an iteration $i \in [I]$, by calling Algorithm 3 with appropriate parameters. Let $n \leftarrow m_i$, $\rho \leftarrow \rho$, and initialize Algorithm 3 with the dataset $\cup_{j \in [J]} \mathcal{D}_i^j$ and $R := \frac{\Delta 4^i}{\lambda_i}$, where

$$\Delta := 3 C_{\mathrm{reg\text{-}pop}} \left( G_k \cdot \left( \frac{\sqrt{d}}{m \sqrt{\rho}} \right)^{1 - \frac{1}{k}} + \frac{G_2}{\sqrt{m}} \right).$$

By Corollary 2, each independent run outputs $x_i^j \in \mathcal{X}$ satisfying, with probability 0.55,

$$\left\| x_i^j - x_i^\star \right\| \le \frac{C_{\text{reg-pop}}}{\lambda_i} \left( G_k \cdot \left( \frac{\sqrt{d}}{m_i \sqrt{\rho}} \right)^{1-\frac{1}{k}} + \frac{G_2}{\sqrt{m_i}} \right) \le \frac{\Delta 4^i}{3\lambda_i} = \frac{R}{3}. \tag{9}$$

Therefore, by a Chernoff bound, with probability $\ge 1 - \frac{\delta}{I}$, at least $0.51J$ of the copies satisfy the above bound, so Fact 2 yields $x_i$ satisfying $\|x_i - x_i^\star\| \le R = \frac{\Delta 4^i}{\lambda_i}$ with the same probability. Union bounding over all $I$ iterations of Algorithm 2 yields the failure probability, and so we obtain the claim from Proposition 2, after plugging in $n = O(m \log(\frac{1}{\delta}))$, since the dominant term is $D\Delta$. The privacy proof follows from the first part of Lemma 1 since for each pair of neighboring databases, exactly one of the datasets $\mathcal{D}_i^j$ are neighboring, and Corollary 2 guarantees privacy of the empirical risk minimization algorithm using that dataset; privacy for all other datasets used is immediate from postprocessing properties of privacy. The gradient complexity comes from aggregating all of the $IJ$ calls to Corollary 2, where we recall the sample sizes decay geometrically.

Finally, if $\frac{1}{\delta}$ is smaller than $\text{polylog}(n)$, for the $i^{\text{th}}$ iteration of Algorithm 2 we instead set $J_i \in [400 \log(\frac{I}{\delta_i}), 500 \log(\frac{I}{\delta_i})]$ where $\delta_i := \frac{\delta}{2^i}$. Then we subdivide a consecutive batch of $\frac{n}{2^i}$ samples into $J_i$ portions, and follow the above proof. It is straightforward to check that (9) still holds with the new value of $m_i = \lfloor \frac{n}{2^i J_i} \rfloor$ because the $4^i$ factor growth on the right-hand side continues to outweigh the change in $m_i$. The error bound follows from Proposition 2, and the privacy proof is identical. □

### A.3 Strongly convex heavy-tailed private SCO via localization

Finally, by following the template of standard localization reductions in the literature (see e.g. Theorem 5.1, [FKT20] or Lemma 5.5, [KLL21]), Theorem 1 obtains an improved rate when all sample functions are strongly convex. For completeness, we state this result below.

**Corollary 3.** *In the setting of Theorem 1, suppose $f(x; s)$ is $\mu$-strongly convex for all $s \in \mathcal{S}$. There is an algorithm which draws $\mathcal{D} \sim \mathcal{P}^n$, queries $C_{\text{sco}} \max(n^2, \frac{n^3 \rho}{d})$ sample gradients (using samples in $\mathcal{D}$) for a universal constant $C_{\text{sco}}$, and outputs $x \in \mathcal{X}$ satisfying, with probability $\ge 1 - \delta$,*

$$F_\mathcal{P}(x) - F_\mathcal{P}(x^\star) \le C_{\text{sco}} \left( \frac{G_k^2}{\mu} \cdot \left( \frac{d \log^3 \left( \frac{1}{\delta} \right)}{n^2 \rho} \right)^{1-\frac{1}{k}} + \frac{G_2^2}{\mu} \cdot \frac{\log \left( \frac{1}{\delta} \right)}{n} \right).$$

*Proof.* This is immediate from the development in Section 5.1 (and the proof of Theorem 5.1) of [FKT20], but we mention one slight difference here. Our guarantees in Theorem 1 do not scale with the initial distance bound to the function minimizer, and instead scale with the domain size, which makes it less directly compatible with the standard localization framework in [FKT20]. However, because Theorem 1 holds with high probability, we also have explicit bounds on the domain size via function error, as seen in the proof of Theorem 5.1 in [FKT20], so we can explicitly truncate our domain to have smaller domain without removing the minimizer. With this modification, the claim follows directly from Theorem 5.1 in [FKT20]. □

## B  Optimal Algorithms in the Known Lipschitz Setting

Compared to the standard Lipschitz setting (i.e. the $\infty$-heavy-tailed private SCO problem), our algorithm in Section 3 has two downsides: it pays a polylogarithmic overhead in the utility, and it requires an extra aggregation step. In this section, assuming we are in the *known Lipschitz $k$-heavy-tailed* setting (see the following Assumption 2, and Definition 4), we provide a simple reduction to the standard Lipschitz setting, resulting in optimal rates.

**Assumption 2** (Known Lipschitz $k$-heavy-tailed distributions). *In the setting of Assumption 1, suppose that for each $s \in \mathcal{S}$ we know a value $\overline{L}_s \ge L_s$. For $k \in \mathbb{N}$ satisfying $k \ge 2$, we say $\mathcal{P}$ satisfies the* known Lipschitz $k$-heavy tailed assumption *if, for a sequence of monotonically nondecreasing $\{G_j\}_{j \in [k]}$, we have $\mathbb{E}_{s \sim \mathcal{P}}[\overline{L}_s^j] \le G_j^j < \infty$ for all $j \in [k]$.*

Note that Assumption 2 clearly implies Assumption 1, but gives us additional access to Lipschitz overestimates with bounded moments. We require some additional definitions used throughout the section. First, we augment $\mathcal{S}$ with a designated element $s_0 \notin \mathcal{S}$, and define

$$f(x; s_0) = 0 \text{ for all } x \in \mathcal{X}. \tag{10}$$

We also define a truncated distribution parameterized by $C \geq 0$, where we use $f(\cdot; s_0)$ in place of sample functions with large Lipschitz overestimates, following notation of Assumption 2:

$$f^C(x; s) := \begin{cases} f(x; s) & \overline{L}_s \leq C \\ f(x; s_0) & \overline{L}_s > C \end{cases}, \; f^C(x; s_0) := f(x; s_0), \; F_{\mathcal{P}}^C(x; s) := \mathbb{E}_{s \sim \mathcal{P}} \left[ f^C(x; s) \right]. \tag{11}$$

We denote $\mathcal{S}_0 := \mathcal{S} \cup \{s_0\}$, and for $\mathcal{D} \in \mathcal{S}^n$, the dataset $\mathcal{D}^C \in \mathcal{S}_0^n$ replaces all $s \in \mathcal{D}$ satisfying $\overline{L}_s > C$ with $s_0$. We additionally provide a second reduction in the known Lipschitz heavy-tailed setting, when all sample functions are assumed to be $\mu$-strongly convex. Because our treatments of these cases are slightly different, we use different notation when $\mu = 0$ and $\mu > 0$, for convenience of exposition. Fixing an arbitrary point $\bar{x} \in \mathcal{X}$, for $\mu > 0$, instead of using the constant 0 function as in (10), we define a strongly convex alternative $f(\cdot; s_\mu)$, for a designated element $s_\mu$:

$$f(x; s_\mu) = \frac{\mu}{2} \|x - \bar{x}\|^2, \text{ for all } x \in \mathcal{X}. \tag{12}$$

The truncated distribution parameterized by $C \geq \mu D$, is defined in a similar way:

$$f_\mu^C(x; s) := \begin{cases} f(x; s) & \overline{L}_s \leq C \\ f(x; s_\mu) & \overline{L}_s > C \end{cases}, \; f_\mu^C(x; s_\mu) := f(x; s_\mu), \; F_{\mathcal{P}}^{C,\mu}(x; s) := \mathbb{E}_{s \sim \mathcal{P}} \left[ f_\mu^C(x; s) \right]. \tag{13}$$

We denote $\mathcal{S}_\mu := \mathcal{S} \cup \{s_\mu\}$, and for $\mathcal{D} \in \mathcal{S}^n$, the dataset $\mathcal{D}_\mu^C \in \mathcal{S}_\mu^n$ replaces every $s \in \mathcal{D}$ such that $\overline{L}_s > C$ with $s_\mu$. Our focus on the regime $C \geq \frac{\mu D}{4}$ is motivated by the following well-known claim.

**Lemma 6.** *Let $\mathcal{X} \subseteq \mathbb{R}^d$ be compact and convex satisfying $\mathsf{diam}(\mathcal{X}) = D$, and suppose $f : \mathcal{X} \to \mathbb{R}$ is $L$-Lipschitz and $\mu$-strongly convex. Then, $L \geq \frac{\mu D}{4}$.*

*Proof.* Let $x^\star := \mathrm{argmin}_{x \in \mathcal{X}} f(x)$. By strong convexity, for all $x \in \mathcal{X}$,

$$\frac{\mu}{2} \|x - x^\star\|^2 \leq f(x) - f(x^\star) \leq L \|x - x^\star\|.$$

Now, choose $x$ such that $\|x - x^\star\| \geq \frac{D}{2}$. To see this is always possible, let $x, x' \in \mathcal{X}$ realize $\|x - x'\| = D$; then at least one of $x, x'$ must have distance $\geq \frac{D}{2}$ from $x^\star$ by the triangle inequality. The conclusion follows by rearranging after using our choice of $x$. $\square$

In other words, if $C < \frac{\mu D}{4}$ then no sample function will survive the truncation in (13). Finally, we parameterize the performance of algorithms in the standard Lipschitz setting.

**Definition 5** (Lipschitz private SCO algorithm). *We say $\mathcal{A}$ is an $L$-Lipschitz private SCO algorithm if it takes input $(\mathcal{D}, \rho, \mathcal{X})$, where $\mathcal{D} \in \mathcal{S}^n$ is drawn i.i.d. from $\mathcal{P}$, a distribution over $\mathcal{S}$ where every $s \in \mathcal{S}$ induces $L$-Lipschitz $f(\cdot; s)$ over $\mathcal{X} \subset \mathbb{R}^d$, $\mathcal{A}(\mathcal{D}, \rho, \mathcal{X}) \in \mathcal{X}$, and $\mathcal{A}$ satisfies $\rho$-CDP. We denote*

$$\mathsf{Err}(\mathcal{A}) := \mathbb{E}_{\mathcal{A}} \left[ F_{\mathcal{P}} \left( \mathcal{A}(\mathcal{D}, \rho, \mathcal{X}) \right) \right] - \min_{x \in \mathcal{X}} F_{\mathcal{P}}(x),$$

*where $F_{\mathcal{P}}(x) := \mathbb{E}_{s \sim \mathcal{P}} f(x; s)$, and denote the number of sample gradients queried by $\mathcal{A}$ by $\mathsf{N}(\mathcal{A})$. Moreover, if each Lipschitz function $f(; s)$ is $\mu$-strongly convex over the convex domain $\mathcal{X}$, we say $\mathcal{A}$ is an $L$-Lipschitz, $\mu$-strongly convex private SCO algorithm, and define $\mathsf{Err}(\mathcal{A})$, $\mathsf{N}(\mathcal{A})$ as before.*

With this notation in place, we state our reduction.

We begin with a simple bound relating $F_{\mathcal{P}}^C$, $F_{\mathcal{P}}^{C,\mu}$ and $F_{\mathcal{P}}$.

**Lemma 7.** *Let $F_{\mathcal{P}}$ be defined as in Definition 4, where $\mathcal{P}$ satisfies Assumption 2, and define $F_{\mathcal{P}}^C$ as in (11). Then, $F_{\mathcal{P}} - F_{\mathcal{P}}^C$ is $\frac{G_k^k}{(k-1)C^{k-1}}$-Lipschitz, and $F_{\mathcal{P}} - F_{\mathcal{P}}^{C,\mu}$ is $\frac{G_k^k}{(k-1)C^{k-1}} + \frac{4G_k^{k+1}}{C^k}$-Lipschitz.*

---

**Algorithm 4:** KnownLipReduction$(\mathcal{D}, C, \mu, \rho, \mathcal{X}, \mathcal{A})$

---

1 **Input:** Dataset $\mathcal{D} \in \mathcal{S}^n$, clip threshold $C \in \mathbb{R}_{\geq 0}$, strong convexity parameter $\mu \in \mathbb{R}_{\geq 0}$, privacy parameter $\rho \in \mathbb{R}_{>0}$, domain $\mathcal{X} \in \mathbb{R}^d$, $C$-Lipschitz private SCO algorithm $\mathcal{A}$ (if $\mu = 0$), or $C$-Lipschitz $\mu$-strongly convex private SCO algorithm $\mathcal{A}$ (if $\mu > 0$)

2 **if** $\mu = 0$ **then**

3 | **Return:** $\mathcal{A}(\mathcal{D}^C, \rho, \mathcal{X})$

4 **end**

5 **else**

6 | **Return:** $\mathcal{A}(\mathcal{D}^C_\mu, \rho, \mathcal{X})$

7 **end**

---

*Proof.* For $s \in \mathcal{S}$, let $\pi(s) := s_0$ if $\overline{L}_s > C$, and otherwise let $\pi(s) := s$. For any $x, x' \in \mathcal{X}$, we have

$$\left(F_\mathcal{P}(x) - F_\mathcal{P}^C(x)\right) - \left(F_\mathcal{P}(x') - F_\mathcal{P}^C(x')\right) = \mathbb{E}_{s \sim \mathcal{P}}\left[f(x; s) - f(x; \pi(s)) - f(x'; s) + f(x'; \pi(s))\right]$$
$$= \mathbb{E}_{s \sim \mathcal{P}}\left[(f(x; s) - f(x'; s))\, \mathbb{I}_{\overline{L}_s > C}\right]$$
$$\leq \mathbb{E}_{s \sim \mathcal{P}}\left[\overline{L}_s \,\|x - x'\|\, \mathbb{I}_{\overline{L}_s > C}\right] \leq \frac{G_k^k}{(k-1)C^{k-1}} \,\|x - x'\|.$$

In the second line, we used that $\pi(s) = s$ unless $\overline{L}_s > C$, in which case $f(\cdot; \pi(s)) = 0$ uniformly. The last line used the definition of $\overline{L}_s$ and Fact 1 with $X \leftarrow \overline{L}_s$, recalling Assumption 2.

Next, we analyze $F_\mathcal{P}^{C,\mu}$. Overloading $\pi(s) := s_\mu$ if $\overline{L}_s > C$, and letting $\pi(s) := s$ otherwise,

$$\left(F_\mathcal{P}(x) - F_\mathcal{P}^{C,\mu}(x)\right) - \left(F_\mathcal{P}(x') - F_\mathcal{P}^{C,\mu}(x')\right)$$
$$= \mathbb{E}_{s \sim \mathcal{P}}\left[f(x; s) - f(x; \pi(s)) - f(x'; s) + f(x'; \pi(s))\right]$$
$$= \mathbb{E}_{s \sim \mathcal{P}}\left[(f(x; s) - f(x'; s) + f(x'; s_\mu) - f(x; s_\mu))\, \mathbb{I}_{\overline{L}_s > C}\right]$$
$$\leq \mathbb{E}_{s \sim \mathcal{P}}\left[\overline{L}_s \,\|x - x'\|\, \mathbb{I}_{\overline{L}_s > C}\right] + \mathbb{E}_{s \sim \mathcal{P}}\left[4G_k \,\|x - x'\|\, \mathbb{I}_{\overline{L}_s > C}\right]$$
$$\leq \left(\frac{G_k^k}{(k-1)C^{k-1}} + \frac{4G_k^{k+1}}{C^k}\right) \|x - x'\|.$$

In the third line, we used that $\mu D \leq 4G_1 \leq 4G_k$ by Lemma 6 and Lemma 2 to show that $f(\cdot; s_\mu)$ is $4G_k$-Lipschitz over $\mathcal{X}$. Finally, the last line used Markov's inequality to bound $\mathbb{E}[\mathbb{I}_{\overline{L}_s > C}]$. $\qquad\square$

Using Lemma 7, we provide a straightforward analysis of Algorithm 4.

**Proposition 3.** *Consider an instance of known-Lipschitz $k$-heavy-tailed private SCO (Definition 4), and let $\rho \geq 0$. If $\mathcal{A}$ is a $C$-Lipschitz private SCO algorithm (Definition 5) and $\mu = 0$, Algorithm 4 using $\mathcal{A}$ is a $\rho$-CDP algorithm which outputs $x \in \mathcal{X}$ satisfying*

$$\mathbb{E}\left[F_\mathcal{P}(x) - F_\mathcal{P}(x^\star)\right] \leq \mathsf{Err}(\mathcal{A}) + \frac{G_k^k D}{(k-1)C^{k-1}}, \text{ where } x^\star := \operatorname*{argmin}_{x \in \mathcal{X}} F_\mathcal{P}(x).$$

*Further, if $f(\cdot; s)$ is $\mu$-strongly convex for all $s \in \mathcal{S}$ and $\mathcal{A}$ is a $C$-Lipschitz, $\mu$-strongly convex private SCO algorithm for $\mu > 0$, Algorithm 4 using $\mathcal{A}$ is a $\rho$-CDP algorithm which outputs $x \in \mathcal{X}$ satisfying*

$$\mathbb{E}\left[F_\mathcal{P}(x) - F_\mathcal{P}(x^\star)\right] \leq \mathsf{Err}(\mathcal{A})$$
$$+ \left(\frac{G_k^k}{(k-1)C^{k-1}} + \frac{4G_k^{k+1}}{C^k}\right)\left(\frac{2G_k^k}{\mu(k-1)C^{k-1}} + \frac{8G_k^{k+1}}{\mu C^k} + \sqrt{\frac{2}{\mu} \cdot \mathsf{Err}(\mathcal{A})}\right),$$
$$\text{where } x^\star := \operatorname*{argmin}_{x \in \mathcal{X}} F_\mathcal{P}(x).$$

*In either case, Algorithm 4 queries $\mathsf{N}(\mathcal{A})$ sample gradients (using samples in $\mathcal{D}$).*

*Proof.* For the first utility claim, letting $x^{\star,C} := \arg\min_{x \in \mathcal{X}} F_{\mathcal{P}}^C(x)$, we have

$$
\begin{aligned}
\mathbb{E}\left[F_{\mathcal{P}}(x) - F_{\mathcal{P}}(x^\star)\right] &= \mathbb{E}\left[F_{\mathcal{P}}^C(x) - F_{\mathcal{P}}^C(x^\star)\right] \\
&\quad + \mathbb{E}\left[\left(F_{\mathcal{P}}(x) - F_{\mathcal{P}}^C(x)\right) - \left(F_{\mathcal{P}}(x^\star) - F_{\mathcal{P}}^C(x^\star)\right)\right] \\
&\leq \mathbb{E}\left[F_{\mathcal{P}}^C(x) - F_{\mathcal{P}}^C(x^{\star,C})\right] + \frac{G_k^k}{(k-1)C^{k-1}}\mathbb{E}\left[\|x - x^\star\|\right] \qquad (14) \\
&\leq \mathsf{Err}(\mathcal{A}) + \frac{G_k^k D}{(k-1)C^{k-1}},
\end{aligned}
$$

where the first inequality used the definition of $x^{\star,C}$ and Lemma 7, and the second used the definition of $\mathsf{Err}$ and $\mathsf{diam}(\mathcal{X}) = D$. For the second claim, we first have

$$
\mathbb{E}\left[\frac{\mu}{2}\left\|x - x^{\star,C}\right\|^2\right] \leq \mathsf{Err}(\mathcal{A})
$$

by the definition of $\mathsf{Err}(\mathcal{A})$ and $\mu$-strong convexity of $F_{\mathcal{P}}^{C,\mu}$, so that $\mathbb{E}[\|x - x^{\star,C}\|] \leq \left(\frac{2}{\mu}\mathsf{Err}(\mathcal{A})\right)^{1/2}$ by Jensen's inequality. Moreover, we also have

$$
\begin{aligned}
\frac{\mu}{2}\left\|x^{\star,C} - x^\star\right\|^2 &\leq F_{\mathcal{P}}(x^{\star,C}) - F_{\mathcal{P}}(x^\star) \\
&\leq \left(F_{\mathcal{P}}(x^{\star,C}) - F_{\mathcal{P}}^C(x^{\star,C})\right) - \left(F_{\mathcal{P}}(x^\star) - F_{\mathcal{P}}^C(x^\star)\right) \\
&\leq \left(\frac{G_k^k}{(k-1)C^{k-1}} + \frac{4G_k^{k+1}}{C^k}\right)\left\|x^{\star,C} - x^\star\right\|,
\end{aligned}
$$

where we use optimality of $x^{\star,C}$ in the second inequality, and Lemma 7 in the third. Combining,

$$
\mathbb{E}\left\|x - x^\star\right\| \leq \frac{2}{\mu} \cdot \left(\frac{G_k^k}{(k-1)C^{k-1}} + \frac{4G_k^{k+1}}{C^k}\right) + \sqrt{\frac{2}{\mu} \cdot \mathsf{Err}(\mathcal{A})},
$$

and then the claim follows by substituting this bound into (14). $\qquad\square$

We can use any existing optimal algorithms for DP-SCO to instantiate our reduction. In particular, we can use the algorithm of [FKT20], denoted by $\mathcal{A}_{\mathsf{Lip}}$, which has the following guarantees. For simplicity of exposition, we focus on the case where our functions do not possess additional regularity properties e.g. smoothness, and we also focus on the simplest $\mathcal{A}_{\mathsf{Lip}}$ which attains the optimal utility bound. Because of the generality of our reduction, however, improvements can be made by using more structured or faster subroutines as $\mathcal{A}_{\mathsf{Lip}}$, such as the smooth DP-SCO algorithms of [FKT20] or the Lipschitz DP-SCO algorithms of e.g. [AFKT21, KLL21, CJJ$^+$23], which are more query-efficient, sometimes at the cost of logarithmic factors in the utility (in the case of [CJJ$^+$23]).

**Proposition 4.** *Let $\mathcal{P}$ be a distribution over $\mathcal{S}$ such that $f(\cdot; s)$ is $L$-Lipschitz and convex for all $s \in \mathcal{S}$. There exists a constant $C_{\mathsf{Lip}}$ such that given $\mathcal{D} \sim \mathcal{S}^n$, the algorithm $\mathcal{A}_{\mathsf{Lip}}$ is $\rho$-CDP and outputs $x_{\mathrm{priv}}$ such that, for a universal constant $C_{\mathsf{Lip}}$, letting $x^\star := \arg\min_{x \in \mathcal{X}} F_{\mathcal{P}}(x)$,*

$$
\mathbb{E}[F_{\mathcal{P}}(x_{\mathrm{priv}}) - F_{\mathcal{P}}(x^\star)] \leq C_{\mathsf{Lip}} \cdot \left(\frac{G_2 D}{\sqrt{n}} + \frac{L D \sqrt{d}}{n\sqrt{\rho}}\right),
$$

*and $\mathcal{A}_{\mathsf{Lip}}$ queries $\leq C_{\mathsf{Lip}} \max(n^2, \frac{n^3 \rho}{d})$ sample gradients (using samples in $\mathcal{D}$), where $G_2$ is defined as in Assumption 1. Moreover, if $f(\cdot; s)$ is $\mu$-strongly convex for all $s \in \mathcal{S}$, then*

$$
\mathbb{E}[F_{\mathcal{P}}(x_{\mathrm{priv}}) - F_{\mathcal{P}}(x^\star)] \leq C_{\mathsf{Lip}} \cdot \left(\frac{G_2^2}{\mu n} + \frac{L^2 d}{\mu n^2 \rho}\right),
$$

*and $\mathcal{A}_{\mathsf{Lip}}$ queries $\leq C_{\mathsf{Lip}} \max(n^2, \frac{n^3 \rho}{d})$ sample gradients (using samples in $\mathcal{D}$).*

*Proof.* This follows from developments in [FKT20], but we briefly explain any discrepancies. The $\mu = 0$ case applies Theorem 4.8 in [FKT20], where for simplicity we consider the full-batch variant

which does not subsample.[8] Moreover, Theorem 4.8 in [FKT20] is stated with a dependence on $L$ rather than $G_2$ on the $n^{-1/2}$ term, but inspecting the proof shows it only uses a second moment bound. The $\mu > 0$ case follows from Theorem 5.1 of [FKT20], using Theorem 4.8 as a subroutine. $\qquad \square$

We are now ready to present our main result in this section, using our reduction with $\mathcal{A}_{\mathsf{Lip}}$.

**Theorem 2.** *Consider an instance of known-Lipschitz $k$-heavy-tailed private SCO (Definition 4), let $\rho \geq 0$, and let $x^\star := \arg\min_{x \in \mathcal{X}} F_{\mathcal{P}}(x)$. Algorithm 4 with $C \leftarrow G_k(\frac{n\sqrt{\rho}}{\sqrt{d}})^{\frac{1}{k}}$ using $\mathcal{A}_{\mathsf{Lip}}$ in Proposition 4 is a $\rho$-CDP algorithm which outputs $x \in \mathcal{X}$ satisfying, for a universal constant $C_{\mathsf{HT}}$,*

$$\mathbb{E}\left[F_{\mathcal{P}}(x) - F_{\mathcal{P}}(x^\star)\right] \leq C_{\mathsf{HT}}\left(\frac{G_2 D}{\sqrt{n}} + G_k D \cdot \left(\frac{\sqrt{d}}{n\sqrt{\rho}}\right)^{1-\frac{1}{k}}\right),$$

*querying $\leq C_{\mathsf{HT}} \max(n^2, \frac{n^3 \rho}{d})$ sample gradients (using samples in $\mathcal{D}$). Further, if $f(\cdot; s)$ is $\mu$-strongly convex for all $s \in \mathcal{S}$, Algorithm 4 with $C \leftarrow G_k(\frac{n^2 \rho}{d})^{\frac{1}{2k}}$ using $\mathcal{A}_{\mathsf{Lip}}$ in Proposition 4 is a $\rho$-CDP algorithm which outputs $x \in \mathcal{X}$ satisfying*

$$\mathbb{E}\left[F_{\mathcal{P}}(x) - F_{\mathcal{P}}(x^\star)\right] \leq C_{\mathsf{HT}}\left(\frac{G_2^2}{\mu n} + \frac{G_k^2}{\mu} \cdot \left(\frac{d}{n^2 \rho}\right)^{1-\frac{1}{k}}\right),$$

*querying $\leq C_{\mathsf{HT}} \max(n^2, \frac{n^3 \rho}{d})$ sample gradients (using samples in $\mathcal{D}$).*

*Proof.* Throughout the proof, assume without loss of generality that $d \leq n^2 \rho$, as otherwise all stated bounds are vacuous since the additive function value range over $\mathcal{X}$ is at most $G_1 D \leq \frac{4G_1^2}{\mu}$ by Lemma 6 and Lemma 2. This also implies that $C \geq G_k$ in either case.

In the $\mu = 0$ case, Proposition 3 and the guarantees of $\mathcal{A}_{\mathsf{Lip}}$ in Proposition 4 imply that

$$\mathbb{E}\left[F_{\mathcal{P}}(\hat{x}) - F_{\mathcal{P}}(x^\star)\right] \leq \mathsf{Err}(\mathcal{A}_{\mathsf{Lip}}) + \frac{G_k^k D}{C^{k-1}}$$

$$\leq C_{\mathsf{Lip}} \cdot \left(\frac{G_2 D}{\sqrt{n}} + \frac{CD\sqrt{d}}{n\sqrt{\rho}}\right) + \frac{G_k^k D}{C^{k-1}}$$

$$\leq (C_{\mathsf{Lip}} + 2)\left(\frac{G_2 D}{\sqrt{n}} + G_k D \cdot \left(\frac{\sqrt{d}}{n\sqrt{\rho}}\right)^{1-\frac{1}{k}}\right),$$

where the last inequality follows from our choice of $C$. Next, we consider $\mu > 0$. Proposition 3 and the guarantees of $\mathcal{A}_{\mathsf{Lip}}$ in Proposition 4 for this case imply that, assuming $C_{\mathsf{Lip}} \geq 2$ without loss,

$$\mathbb{E}\left[F_{\mathcal{P}}(\hat{x}) - F_{\mathcal{P}}(x^\star)\right] \leq \mathsf{Err}(\mathcal{A}_{\mathsf{Lip}}) + \frac{5G_k^k}{C^{k-1}}\left(\sqrt{\frac{2\mathsf{Err}(n, d, \rho, C, D)}{\mu}} + \frac{10G_k^k}{\mu C^{k-1}}\right)$$

$$\leq C_{\mathsf{Lip}} \cdot \left(\frac{G_2^2}{\mu n} + \frac{C^2 d}{\mu n^2 \rho} + \frac{5G_k^k}{C^{k-1}}\left(\frac{G_2}{\mu\sqrt{n}} + \frac{C\sqrt{d}}{\mu n\sqrt{\rho}} + \frac{10G_k^k}{\mu C^{k-1}}\right)\right)$$

$$\leq (C_{\mathsf{Lip}} + 61) \cdot \left(\frac{G_2^2}{\mu n} + \frac{G_k^2}{\mu} \cdot \left(\frac{d}{n^2 \rho}\right)^{1-\frac{1}{k}}\right),$$

where we used $C \geq G_k$ to simplify bounds, and applied our choice of $C$. $\qquad \square$

---

[8] The subsampled variant only satisfies a weaker variant of CDP called truncated CDP, with an upside of using $n$ times fewer sample gradient queries, but this is less comparable to the lower bounds in [LR23].

## C Fast Algorithms for Smooth Functions

In this section, we develop a linear-time algorithm for the smooth setting where we additionally assume $f(\cdot; s)$ is $\beta$-smooth for all $s \in \mathcal{S}$. Our algorithm attains nearly-optimal rates for a sufficiently small value of $\beta$, and is based on the localization framework of [FKT20]. To apply this framework, we show that a variant of clipped DP-SGD (see Algorithm 5) is stable in the heavy-tailed setting with high probability. We then ensure that stability holds for any input dataset (not necessarily sampled from a distribution $P$), by using the sparse vector technique [DR14] to verify that the number of clipped gradients is not too large. In Appendix C.1, we provide some standard preliminary results from the literature. We use these results in Appendix C.2, where we state our algorithm in full as Algorithm 7 and analyze it in Theorem 3, the main result of this section.

### C.1 Helper tools

First, we state a standard bound on the contractivity of smooth gradient descent iterations.

**Fact 3** (Lemma 3.7, [HRS16])**.** *Let $f : \mathcal{X} \to \mathbb{R}$ be $\beta$-smooth, and let $\eta \leq \frac{2}{\beta}$. Then for any $x, x' \in \mathcal{X}$,*

$$\|(x - x') - \eta(\nabla f(x) - \nabla f(x'))\| \leq \|x - x'\|.$$

Next, we provide a standard utility bound on a one-pass SGD algorithm using clipped gradients.

---

**Algorithm 5:** OnePass-Clipped-SGD$(\mathcal{D}, C, \eta, T, \mathcal{X}, x_0)$

---

1 **Input:** Dataset $\mathcal{D} = \{s_t\}_{t \in [T]} \in \mathcal{S}^T$, clip threshold $C \in \mathbb{R}_{\geq 0}$, step size $\eta \in \mathbb{R}_{\geq 0}$, iteration count $T \in \mathbb{N}$, domain $\mathcal{X} \subset \mathbb{B}(x_0, D)$ for $x_0 \in \mathcal{X}$
2 **for** $0 \leq t < T$ **do**
3 $\quad$ $x_{t+1} \leftarrow \text{argmin}_{x \in \mathcal{X}} \{\eta \langle \Pi_C(\nabla f(x_t; s_{t+1})), x \rangle + \frac{1}{2} \|x - x_t\|^2\}$
4 **end**
5 **Return:** $\hat{x} \leftarrow \frac{1}{T} \sum_{0 \leq t < T} x_t$

---

**Lemma 8.** *Consider an instance of $k$-heavy-tailed private SCO, following notation in Definition 4, and let $u \in \mathcal{X}$ be independent of $\mathcal{D}$. Assuming $\mathcal{D} \sim \mathcal{P}^T$ i.i.d., Algorithm 5 outputs $\hat{x} \in \mathcal{X}$ satisfying*

$$\mathbb{E}\left[F_{\mathcal{P}}(\hat{x}) - F_{\mathcal{P}}(u)\right] \leq \frac{\|x_0 - u\|^2}{2\eta T} + \frac{\eta G_2^2}{2} + \frac{G_k^k D}{(k-1)C^{k-1}}.$$

*Proof.* To simplify notation, let $g_t := \nabla f(x_t; s_{t+1})$ for all $0 \leq t < T$, and let $\hat{g}_t := \mathcal{T}_C(g_t)$. Because $s_{t+1} \sim \mathcal{P}$ is independent of $x_t$, we have that $\mathbb{E}g_t = \nabla F_{\mathcal{P}}(x_t)$. Therefore, in iteration $t$,

$$
\begin{aligned}
F_{\mathcal{P}}(x_t) - F_{\mathcal{P}}(u) &= \mathbb{E}\left[\langle g_t, x_t - u \rangle\right] \\
&\leq \mathbb{E}\left[\langle \hat{g}_t, x_t - u \rangle + \|g_t - \hat{g}_t\| D\right] \\
&\leq \mathbb{E}\left[\frac{1}{2}\|x_t - u\|^2 - \frac{1}{2}\|x_{t+1} - u\|^2 + \frac{\eta G_2^2}{2}\right] + \frac{G_k^k D}{(k-1)C^{k-1}},
\end{aligned}
\tag{15}
$$

where all expectations are conditional on the first $t$ iterations of the algorithm, and taken over the randomness of $s_{t+1}$. In the third line, we used the first-order optimality condition on $x_{t+1}$, applied Fact 1 to bound $\mathbb{E}\|g_t - \hat{g}_t\|$, and used

$$\mathbb{E}\|\hat{g}_t\|^2 \leq \mathbb{E}\|g_t\|^2 \leq G_2^2. \tag{16}$$

Summing across all iterations and dividing by $T$ yields the result upon iterating expectations. $\quad\square$

We also note the following straightforward generalization of Lemma 8 to the case of randomized clipping thresholds, which is used in our later development.

**Corollary 4.** *For $C, \hat{C} \geq 0$ and $g \in \mathbb{R}^d$, define the operation*

$$\Pi_{C,\hat{C}}(g) := \begin{cases} \Pi_C(g) & \|g\| \geq \hat{C} \\ g & \text{else} \end{cases}.$$

*If Algorithm 5 is run with $\Pi_C(\nabla f(x_t; s_{t+1}))$ replaced by $\Pi_{C,\hat{C}_t}(\nabla f(x_t; s_{t+1}))$ where $\hat{C}_t$ is independent of $s_{t+1}$ and satisfies $\hat{C}_t \geq \frac{C}{2}$ for all $0 \leq t < T$, then following notation in Lemma 8,*

$$\mathbb{E}\left[F_{\mathcal{P}}(\hat{x}) - F_{\mathcal{P}}(u)\right] \leq \frac{\|x_0 - u\|^2}{2\eta T} + 2\eta G_2^2 + \frac{G_k^k D}{(k-1)(\frac{C}{2})^{k-1}}.$$

*Proof.* For a fixed iteration $0 \leq t < T$, the calculation (16) changes in two ways. First, in place of the variance bound (16) (which used $\|\hat{g}_t\| \leq \|g_t\|$ deterministically), when using the modified clipping operators we require the modified deterministic bound

$$\|\hat{g}_t\| \leq 2\|g_t\|,$$

which follows because $\|\hat{g}_t\| \neq \|g_t\|$ (which implies $C = \|\hat{g}_t\|$) only if $\|g_t\| \geq \frac{C}{2}$. Moreover, in place of the bias bound $\mathbb{E}\|g_t - \hat{g}_t\| \leq \frac{G_k^k}{(k-1)C^{k-1}}$ which followed from Fact 1, we instead have

$$\mathbb{E}\left\|\Pi_{C,\hat{C}_t}(g_t) - g_t\right\| = \mathbb{E}\left[\left|\frac{C}{\|g_t\|} - 1\right| \|g_t\| \mathbb{1}_{\|g_t\| \geq \max(\hat{C}_t, C)}\right] \leq \mathbb{E}\left[\|g_t\| \mathbb{1}_{\|g_t\| \geq \frac{C}{2}}\right] \leq \frac{G_k^k}{(k-1)(\frac{C}{2})^{k-1}}.$$

The conclusion follows by adjusting these constants appropriately in Lemma 8. $\qquad\square$

Next, for $R, \tau \geq 0$, we let $\mathrm{BLap}(R, \tau)$ denote the bounded Laplace distribution with scale parameter $R$ and truncation threshold $\tau$ be defined as the conditional distribution of $\xi \sim \mathrm{Lap}(R)$ on the event $|\xi| \leq \tau$ (recall that $\mathrm{Lap}(R)$ has a density function $\propto \exp(-\frac{1}{R}|\xi|)$). It is a standard calculation that

$$\Pr_{\xi \sim \mathrm{Lap}(R)}\left[|\xi| \leq R \log\left(\frac{1}{\delta}\right)\right] = 1 - \delta, \tag{17}$$

so that the total variation distance between $\mathrm{Lap}(R)$ and $\mathrm{BLap}(R, R\log(\frac{1}{\delta}))$ is $\delta$. We hence have the following bounded generalization of the privacy given by the Laplace mechanism.

**Lemma 9.** *Let $\varepsilon, \delta \in (0, 1)$. If $S(\mathcal{D}) \in \mathbb{R}$ is a $\Delta$-sensitive statistic of the dataset $\mathcal{D}$, i.e. for neighboring datasets $\mathcal{D}, \mathcal{D}'$ we have that $|S(\mathcal{D}) - S(\mathcal{D}')| \leq \Delta$, then the bounded Laplace mechanism which outputs $S(\mathcal{D}) + \xi$ where $\xi \sim \mathrm{BLap}(\frac{\Delta}{\varepsilon}, \tau)$ for any $\tau \geq \frac{\Delta}{\varepsilon}\log(\frac{4}{\delta})$ satisfies $(\varepsilon, \delta)$-DP.*

*Proof.* For notational simplicity, let $\mathcal{A}$ denote the Laplace mechanism (which samples $\xi \sim \mathrm{Lap}(\frac{\Delta}{\varepsilon})$ instead of $\mathrm{BLap}(\frac{\Delta}{\varepsilon}, \tau)$), let $\overline{\mathcal{A}}$ denote the bounded Laplace mechanism, and let $\mathcal{E} \subseteq \mathbb{R}$ be an event in the outcome space. By standard guarantees on $(\varepsilon, 0)$-DP of $\mathcal{A}$ (e.g. Theorem 3.6, [DR14]),

$$\begin{aligned}
\Pr\left[\overline{\mathcal{A}}(\mathcal{D}) \in \mathcal{E}\right] &\leq \Pr\left[\mathcal{A}(\mathcal{D}) \in \mathcal{E}\right] + \frac{\delta}{4} \\
&\leq \exp(\varepsilon)\Pr\left[\mathcal{A}(\mathcal{D}') \in \mathcal{E}\right] + \frac{\delta}{4} \leq \exp(\varepsilon)\Pr\left[\overline{\mathcal{A}}(\mathcal{D}') \in \mathcal{E}\right] + \delta,
\end{aligned} \tag{18}$$

for any neighboring datasets, where we used $\exp(\varepsilon) \leq 3$ and that the total variation distance between $(\mathcal{A}(\mathcal{D}), \overline{\mathcal{A}}(\mathcal{D}))$ and $(\mathcal{A}(\mathcal{D}'), \overline{\mathcal{A}}(\mathcal{D}'))$ are bounded by $\frac{\delta}{4}$ by (17). $\qquad\square$

We also use the sparse vector technique (SVT) [DR14], which has been used recently in private optimization in the user-level setting [AL24]. Given an input dataset $\mathcal{D} = \{s_i\}_{i\in[n]} \in \mathcal{S}^n$, SVT takes a stream of queries $q_1, q_2, \ldots, q_T : \mathcal{D} \to \mathbb{R}$ in an online manner. We assume each $q_i$ is $\Delta$-sensitive, i.e. $|q_i(\mathcal{D}) - q_i(\mathcal{D}')| \leq \Delta$ for neighboring datasets $\mathcal{D}, \mathcal{D}' \in \mathcal{S}^n$. One notable difference is that our SVT algorithm will use the bounded Laplace mechanism, rather than the Laplace mechanism, but this distinction is handled similarly to Lemma 9. We provide a guarantee on this variant of SVT in Lemma 10, and pseudocode is provided as Algorithm 6.

**Lemma 10.** *Let $\delta, \varepsilon \in (0, 1)$ and suppose*

$$R \geq \frac{6\Delta}{\varepsilon}\sqrt{c\log\left(\frac{5}{\delta}\right)}, \quad \tau \geq R\log\left(\frac{10T}{\delta}\right) \tag{19}$$

*Algorithm 6 outputs a sequence of answers $\{a_i \in \{\bot, \top\}\}_{i\in[k]}$ for some $k \in [T]$, and is $(\varepsilon, \delta)$-DP.*

---

**Algorithm 6:** $\mathsf{SVT}(\mathcal{D}, \{q_i\}_{i \in [T]}, c, L, R, \tau)$

---

1 **Input:** Dataset $\mathcal{D} = \{s_t\}_{t \in [n]} \in \mathcal{S}^n$, $\Delta$-sensitive queries $\{q_i : \mathcal{S}^n \to \mathbb{R}\}_{i \in [T]}$, count threshold $c \in \mathbb{N}$, query threshold $L \in \mathbb{R}$, scale parameter $R \in \mathbb{R}_{\geq 0}$, truncation threshold $\tau \in \mathbb{R}_{\geq 0}$
2 $i \leftarrow 1$, count $\leftarrow 0$
3 $b \leftarrow L + \xi$ for $\xi \sim \mathrm{BLap}(R, \tau)$
4 **while** $i \in [T]$ **and** count $< c$ **do**
5 $\quad$ $\xi \sim \mathrm{BLap}(2R, 2\tau)$
6 $\quad$ **if** $q_i(\mathcal{D}) + \xi < b$ **then**
7 $\quad\quad$ **Output:** $a_i \leftarrow \perp$
8 $\quad\quad$ $i \leftarrow i + 1$
9 $\quad$ **end**
10 $\quad$ **else**
11 $\quad\quad$ **Output:** $a_i \leftarrow \top$
12 $\quad\quad$ $i \leftarrow i + 1$, count $\leftarrow$ count $+ 1$
13 $\quad\quad$ $b \leftarrow L + \xi$ for $\xi \sim \mathrm{BLap}(R, \tau)$
14 $\quad$ **end**
15 **end**
16 **Halt**

---

*Proof.* The proof is analogous to Lemma 9. Let $\mathcal{A}$ denote SVT run with Laplace noise in place of bounded Laplace noise (i.e. $\tau = \infty$), and let $\overline{\mathcal{A}}$ denote SVT run with bounded Laplace noise. We first claim that $\mathcal{A}$ is $(\varepsilon, \frac{\delta}{5})$-DP, which is immediate from Theorem 3.23 and Theorem 3.20 in [DR14].

Next, by a union bound on all of the $\leq 2T$ random variables sampled, the total variation distance between $(\mathcal{A}(\mathcal{D}), \overline{\mathcal{A}}(\mathcal{D}))$ for any dataset $\mathcal{D}$ is bounded by $\frac{\delta}{5}$. Then, for neighboring datasets $\mathcal{D}, \mathcal{D}'$ and some event $\mathcal{E}$ in the outcome space, repeating the calculation (18),

$$
\begin{aligned}
\Pr\left[\overline{\mathcal{A}}(\mathcal{D}) \in \mathcal{E}\right] &\leq \Pr\left[\mathcal{A}(\mathcal{D}) \in \mathcal{E}\right] + \frac{\delta}{5} \\
&\leq \exp(\varepsilon) \Pr\left[\mathcal{A}(\mathcal{D}') \in \mathcal{E}\right] + \frac{\delta}{5} + \frac{\delta}{5} \leq \exp(\varepsilon) \Pr\left[\overline{\mathcal{A}}(\mathcal{D}') \in \mathcal{E}\right] + \delta.
\end{aligned}
$$

$\square$

## C.2 Algorithm statement and analysis

In this section, we present the full details of our algorithm (see Algorithm 7) and prove its corresponding guarantees, separating out the privacy analysis and utility analysis.

**Algorithm 7:** Localized-Clipped-DP-SGD($\mathcal{D}, x_0, \eta, c, \varepsilon, \delta$)

---

1 **Input:** Dataset $\mathcal{D} \in \mathcal{S}^n$, initial point $x_0 \in \mathcal{X}$, step size $\eta \in \mathbb{R}_{>0}$, parameters $C, c, \omega \in \mathbb{R}_{>0}$, privacy parameters $(\varepsilon, \delta) \in \mathbb{R}_{>0}^2$

2 $I \leftarrow \lfloor \log_2 n \rfloor$, $n \leftarrow 2^I$

3 **for** $i \in [I]$ **do**

4     $n_i \leftarrow \frac{n}{2^i}$, $\eta_i \leftarrow \frac{\eta}{4^i}$, $\omega_i \leftarrow \omega \cdot 6C\eta_i\beta$

5     $\hat{C} \leftarrow C + \mathrm{BLap}(\omega_i, \omega_i \log(\frac{30n_i}{\delta}))$, $\hat{c}_i \leftarrow c + \mathrm{BLap}(\frac{3}{\varepsilon}, \frac{c}{2})$, $\mathrm{count} \leftarrow 0$

6     $x_{i,1} \leftarrow x_{i-1}$

7     **for** $j \in [n_i]$ **do**

8        $s_{i,j} \leftarrow (\sum_{i' \in [i]} n_{i'} + j)^{\mathrm{th}}$ element of $\mathcal{D}$

9        $\nu_{i,j} \sim \mathrm{BLap}(2\omega_i, 2\omega_i \log(\frac{30n_i}{\delta}))$

10        **if** $\|\nabla f(x_{i,j}; s_{i,j})\| + \nu_{i,j} \geq \hat{C}$ **then**

11           $\mathrm{count} \leftarrow \mathrm{count} + 1$

12           $g_{i,j} \leftarrow \Pi_C(\nabla f(x_{i,j}; s_{i,j}))$

13           $\hat{C} \leftarrow C + \mathrm{BLap}(\omega_i, \omega_i \log(\frac{30n_i}{\delta}))$

14        **end**

15        **else**

16           $g_{i,j} \leftarrow \nabla f(x_{i,j}; s_{i,j})$

17        **end**

18        **if** $\mathrm{count} \geq \hat{c}_i$ **then**

19           **Return:** $\perp$

20        **end**

21        $x_{i,j+1} \leftarrow \Pi_{\mathcal{X}}(x_{i,j} - \eta_i g_{i,j})$

22     **end**

23     $\overline{x}_i \leftarrow \frac{1}{n_i} \sum_{j \in [n_i]} x_{i,j}$

24     $x_i \leftarrow \overline{x}_i + \zeta_i$, where $\zeta_i \sim \mathcal{N}(0, \sigma_i^2 \mathbf{I}_d)$ with $\sigma_i = \frac{30C\eta_i\sqrt{\log(3/\delta)}}{\varepsilon}$

25 **end**

26 **Return:** $x_I$

---

The following theorem summarizes the guarantees of Algorithm 7.

**Theorem 3.** *Consider an instance of $k$-heavy-tailed private SCO, following notation in Definition 4, and let $x^\star := \arg\min_{x \in \mathcal{X}} F_{\mathcal{P}}(x)$, and $\varepsilon, \delta \in (0,1)$. Algorithm 7 run with parameters*

$$\eta \leftarrow \min\left(\sqrt{\frac{4}{n}} \cdot \frac{D}{G_2}, \ \frac{DI}{G_k n} \cdot \left(\frac{n^2\varepsilon^2}{14400d\log^2(\frac{15n}{\delta})}\right)^{\frac{k-1}{2k}}\right),$$

$$C \leftarrow 2\left(\frac{G_k^k DIn\varepsilon^2}{14400d\eta \log^2(\frac{15n}{\delta})}\right)^{\frac{1}{k+1}}, \ c \leftarrow \frac{240\sqrt{d}\log(\frac{15n}{\delta})}{\varepsilon}, \ \omega \leftarrow \frac{18}{\varepsilon}\sqrt{2c\log\left(\frac{15}{\delta}\right)},$$

*is $(\varepsilon, \delta)$-DP and outputs $x_I$ that satisfies, for a universal constant $C_{\mathrm{smooth}}$,*

$$\mathbb{E}\left[F_{\mathcal{P}}(x_k) - F_{\mathcal{P}}(x^\star)\right] \leq C_{\mathrm{smooth}}\left(\frac{G_2 D}{\sqrt{n}} + G_k D \cdot \left(\frac{\sqrt{d\log^3(\frac{n}{\delta})}}{n\varepsilon}\right)^{1-\frac{1}{k}}\right),$$

*assuming $f(\cdot; s)$ is $\beta$-smooth for all $s \in \mathcal{S}$, where*

$$\beta \leq \frac{\varepsilon^{1.5}}{24000\eta\sqrt{d}\log^2(\frac{30n}{\delta})}$$

$$= \Theta\left(\max\left(\frac{G_2}{D} \cdot \frac{\sqrt{n}\varepsilon^{1.5}}{\sqrt{d}\log^2(\frac{n}{\delta})}, \ \frac{G_k}{D} \cdot \frac{\varepsilon^{1.5}n}{\sqrt{d}\log(n)\log^2(\frac{n}{\delta})} \cdot \left(\frac{d\log^2(\frac{n}{\delta})}{n^2\varepsilon^2}\right)^{\frac{k-1}{2k}}\right)\right). \tag{20}$$

We now proceed to prove Theorem 3.

**Privacy proof overview.** We first overview the structure of our privacy proof. Consider two neighboring datasets $\mathcal{D}, \mathcal{D}'$ that differ on a single sample $s_{i,j_0} \neq s'_{i,j_0}$. The core argument used to prove privacy is controlling the total number of times when gradients are clipped, so we introduce the variable "count." Note that we have $\|x_{i,j_0+1} - x'_{i,j_0+1}\| = O(C\eta)$ due to the clip operation. If no clip ever happened afterward, then we know $\|x_{i,n_i} - x'_{i,n_i}\| \leq \|x_{i,j_0+1} - x'_{i,j_0+1}\| = O(C\eta)$ due to our smoothness assumption (see Fact 3), which means the algorithm is private. When count is not too large, we can still bound the sensitivity between $\|x_{i,n_i} - x'_{i,n_i}\|$ by $O(C\eta)$. However, when the value of count is larger, there is a risk that the sensitivity of $x_{i,n_i}$ is not bounded as before, and hence we halt the algorithm when count exceeds some appropriate cutoff point $\hat{c}_i$.

One subtle difference between our algorithm and standard uses of SVT is that we add Laplace noise to the cutoff point $c$ to obtain a randomized cutoff $\hat{c}_i$. This is because the sensitivity of the count increment at the $j_0^{\text{th}}$ iteration of phase $i$ is bounded by one, even though $\|\nabla f(x_{i,j_0}; s_{i,j_0})\| - \|\nabla f(x'_{i,j_0}; s'_{i,j_0})\|$ can be arbitrarily large. The guarantees of the bounded Laplace mechanism imply that the noise added in $\hat{c}_i$ hence suffices to privatize count.

In summary, we can control the sensitivity between $\|x_{i,j} - x'_{i,j}\|$ for all $j$ due to the termination condition in Line 18 and our use of bounded Laplace noise, and hence can control the sensitivity of the query for $\|\nabla f(x_{i,j}; s_{i,j})\| - \|\nabla f(x'_{i,j}; s'_{i,j})\|$ for all $j \neq j_0$. By adding Laplace noise on the cutoff $c$, we handle the issue of the sensitivity of the $j_0^{\text{th}}$ query $\|\nabla f(x_{i,j_0}; s_{i,j_0})\|$ being unbounded. If the algorithm succeeds and returns $x_k$, we know the sensitivity $\|x_{i,n_i} - x'_{i,n_i}\|$ is $O(C\eta_i)$ and the privacy guarantee follows from the Gaussian mechanism. If the algorithm fails and outputs $\perp$, the privacy guarantee follows from the bounded Laplace noise on the cutoff point and the guarantees of SVT.

**Privacy proof.** We now provide our formal privacy analysis following this overview. To fix notation in the remainder of the privacy proof, we consider running Algorithm 7 on two neighboring datasets $\mathcal{D}, \mathcal{D}'$ that differ on a single sample $s_{i,j_0} \neq s'_{i,j_0}$, for some $i \in [I]$. By standard postprocessing properties of differential privacy, it suffices to argue that the $i^{\text{th}}$ phase (i.e. the run of the loop in Lines 3 to 25 corresponding to this value of $i$) is private, so we fix $i \in [I]$ in the following discussion.

We let $\{x_{i,j}\}_{j \in [n_i]}$ and $\{x'_{i,j}\}_{j \in [n_i]}$ be the iterates of the $i^{\text{th}}$ phase of Algorithm 7 using $\mathcal{D}$ and $\mathcal{D}'$, and we let $Y_{i,j}$ and $Y'_{i,j}$ be the respective 0-1 indicator variables that count increases by 1 in iteration $j$. We also let $\text{count}_j$ and $\text{count}'_j$ denote the values of count at the end of the $j^{\text{th}}$ iteration, and abusing notation we let $\hat{c}_i, \hat{c}'_i$ be the values of $\hat{c}_i$ in the $i^{\text{th}}$ phase when using $\mathcal{D}$ or $\mathcal{D}'$ respectively. Finally, we denote $\overline{x}_i := \frac{1}{n_i} \sum_{j \in [n_i]} x_{i,j}$ and let $\overline{x}'_i$ denote the average iterate using $\mathcal{D}'$ similarly.

We first bound the sensitivity between the iterates $\{x_{i,j}\}_{j \in [n_i]}$ and $\{x'_{i,j}\}_{j \in [n_i]}$ in the following lemma, assuming $\text{count}_j$ and $\text{count}_{j'}$ are bounded. The proof is deferred to Appendix H.

**Lemma 11.** *Let $t \in [n_i]$, and suppose that $192\eta_i\beta c \leq 1$ and $C \geq 8\omega_i \log(\frac{30n_i}{\delta})$. If $\text{count}_t < \hat{c}_i$, $\text{count}'_t < \hat{c}'_i$, and $Y_{i,j} = Y'_{i,j}$ for all $j < t$ with $j \neq j_0$, then*

$$\|x_{i,t} - x'_{i,t}\| \leq 6C\eta_i.$$

Using this bound on the sensitivity, we are now ready to prove privacy of the algorithm.

**Lemma 12.** *Algorithm 7 is $(\varepsilon, \delta)$-DP if it is run with parameters satisfying*

$$C \geq 8\omega_i \log\left(\frac{30n_i}{\delta}\right), \ c \geq \frac{6}{\varepsilon}\log\left(\frac{12}{\delta}\right), \ \omega \geq \frac{18}{\varepsilon}\sqrt{2c\log\left(\frac{15}{\delta}\right)}, \ 192\eta_i\beta c \leq 1.$$

*Proof.* Recall our assumption that $\mathcal{D}$ and $\mathcal{D}'$ only differ in $s_{i,j_0}$, the $j_0^{\text{th}}$ sample used in the $i^{\text{th}}$ phase of the algorithm. The privacy of all phases of the algorithm other than phase $i$ is immediate from postprocessing properties of DP, so it suffices to argue that phase $i$ is $(\varepsilon, \delta)$-DP. Note also that the conditions of Lemma 11 are met after reparameterizing $\delta \leftarrow \frac{\delta}{4}$. We split our privacy argument into two cases, depending on whether the algorithm terminates on Line 18 or Line 26.

*Termination on Line 18.* We begin with the case where the algorithm outputs $\perp$. We introduce some simplifying notation. For iterations $S \subseteq [n_i]$, define $W_S := \{Y_{i,j}\}_{j \in S}$ to be the 0-1 indicator variables for whether count incremented on iterations $j \in S$ (when run on $\mathcal{D}$), and define $[W]_S :=$

$\sum_{j \in S} Y_{i,j}$ to be their sum. Similarly, define $W'_S$ and $[W']_S$ for when the algorithm is run on $\mathcal{D}'$. Observe that the algorithm outputs $\bot$ iff the following event occurs:

$$Y_{i,j_0} + [W]_{[n_i]\setminus\{j_0\}} \geq \hat{c}_i \iff (Y_{i,j_0} - \hat{c}_i) + [W]_{[n_i]\setminus[j_0]} \geq -[W]_{[j_0-1]}.$$

The right-hand side $-[W]_{[j_0-1]}$ is independent of whether the dataset used was $\mathcal{D}$ or $\mathcal{D}'$, so it suffices to argue about the privacy loss of the random variables $Y_{i,j_0} - \hat{c}_i$ and $W_{[n_i]\setminus[j_0]}$ as a function of the dataset used. First, $Y_{i,j_0} - c$ is clearly a 1-sensitive statistic, so Lemma 9 implies $Y_{i,j_0} - \hat{c}_i$ is $(\frac{\varepsilon}{3}, \frac{\delta}{3})$-indistinguishable as a function of the dataset used. Next, conditioning on the value of $Y_{i,j_0} - \hat{c}_i$, the random variable $W_{[n_i]\setminus[j_0]}$ is an instance of Algorithm 6 run with a fixed threshold $\hat{c}_i - Y_{i,j_0} - [W]_{[j_0-1]} \leq 2c$, where we rename the output variables $\{\bot, \top\}$ to $\{0, 1\}$. Moreover, Lemma 11 and smoothness of each sample function implies that the sensitivity of each query $\|\nabla f(\cdot; s_{i,j})\|$ is bounded by $\Delta := 6C\eta_i\beta$. Therefore, Lemma 10 shows that $W_{[n_i]\setminus[j_0]}$ is $(\frac{\varepsilon}{3}, \frac{\delta}{3})$-indistinguishable, where we note that we adjusted constants appropriately in $\omega$ and the failure probabilities everywhere. By basic composition of DP, this implies $Y_{i,j_0} - \hat{c}_i + [W]_{[n_i]\setminus[j_0]}$ (a postprocessing of $Y_{i,j_0} - \hat{c}_i$ and $W_{[n_i]\setminus[j_0]} \mid Y_{i,j_0} - \hat{c}_i$) is $(\frac{2\varepsilon}{3}, \frac{2\delta}{3})$-DP, as required.

*Termination on Line 26.* Finally, we argue about the privacy when the algorithm does not terminate on Line 18. As before, the sensitivity of $\bar{x}_i$ is bounded by $6C\eta_i$ via Lemma 11 and the triangle inequality, conditioned on a $(\frac{2\varepsilon}{3}, \frac{2\delta}{3})$-indistinguishable event (i.e. the values of $Y_{i,j_0} - \hat{c}_i$ and $W_{[n_i]\setminus[j_0]} \mid Y_{i,j_0} - \hat{c}_i$). Then $x_i$ is $(\frac{\varepsilon}{3}, \frac{\delta}{3})$-indistinguishable by standard bounds on the Gaussian mechanism (Theorem A.1, [DR14]), which completes the proof upon applying basic composition. $\square$

**Utility proof.** The utility proof follows the standard analysis of localized SGD algorithms and a specialized analysis of clipped SGD (Corollary 4). We first state a utility guarantee in each phase.

**Lemma 13.** *Following notation in Algorithm 7, fix $i \in [I]$, and suppose $\mathcal{D} \sim \mathcal{P}^n$ i.i.d. where $\mathcal{P}$ satisfies Assumption 1. For any $x \in \mathcal{X}$, if $C \geq 8\omega_i \log(\frac{30n_i}{\delta})$ and $\frac{c}{4} \geq \max(n \cdot (\frac{2G_k}{C})^k, 6\log(n))$,*

$$\mathbb{E}[F_{\mathcal{P}}(\bar{x}_i) - F_{\mathcal{P}}(x)] \leq \frac{\|x - x_{i-1}\|^2}{2\eta_i n_i} + 2\eta_i G_2^2 + \frac{G_k^k D}{(k-1)(\frac{C}{2})^{k-1}} + \frac{G_2 D}{n^2}.$$

*Proof.* By Markov's inequality, $\mathbb{E}_{s \sim \mathcal{P}}[\mathbb{1}_{L_s > \frac{C}{2}}] \leq (\frac{2G_k}{C})^k$, so the total number of expected samples with $L_s > \frac{C}{2}$ is at most $\frac{c}{4}$. Hence by applying a Chernoff bound,

$$\Pr_{\mathcal{D} \sim \mathcal{P}^n} \left[ \underbrace{\sum_{s \in \mathcal{D}} \mathbb{1}_{L_s > \frac{C}{2}} \leq \frac{c}{2}}_{:= \mathcal{E}} \right] \geq 1 - \frac{1}{n^2}.$$

Conditional on $\mathcal{E}$, the algorithm will not halt (i.e., return $\bot$) and is running one-pass clipped-SGD (Algorithm 5) using the modified clipping operation defined in the precondition in Corollary 4. Then, the statement follows from Corollary 4 as follows: letting $\mathcal{E}^c$ denote the complement of $\mathcal{E}$,

$$\mathbb{E}[F_{\mathcal{P}}(\bar{x}_i) - F_{\mathcal{P}}(x)] = \mathbb{E}[F_{\mathcal{P}}(\bar{x}_i) - F_{\mathcal{P}}(x) \mid \mathcal{E}] \Pr[\mathcal{E}] + \mathbb{E}[F_{\mathcal{P}}(\bar{x}_i) - F_{\mathcal{P}}(x) \mid \mathcal{E}^c] \Pr[\mathcal{E}^c]$$

$$\leq \frac{\|x - x_{i-1}\|^2}{2\eta_i n_i} + 2\eta_i G_2^2 + \frac{G_k^k D}{(k-1)(\frac{C}{2})^{k-1}} + \mathbb{E}[F_{\mathcal{P}}(\bar{x}_i) - F_{\mathcal{P}}(x) \mid \mathcal{E}^c] \Pr[\mathcal{E}^c]$$

$$\leq \frac{\|x - x_{i-1}\|^2}{2\eta_i n_i} + 2\eta_i G_2^2 + \frac{G_k^k D}{(k-1)(\frac{C}{2})^{k-1}} + G_2 D \Pr[\mathcal{E}^c]$$

$$\leq \frac{\|x - x_{i-1}\|^2}{2\eta_i n_i} + 2\eta_i G_2^2 + \frac{G_k^k D}{(k-1)(\frac{C}{2})^{k-1}} + \frac{G_2 D}{n^2},$$

where we used that $F_{\mathcal{P}}$ is $G_1 \leq G_2$-Lipschitz by Lemma 2. $\square$

Combining our privacy and utility guarantees, we are ready to prove this section's main theorem.

*Proof of Theorem 3.* For simplicity, let $\bar{x}_0 := x^\star$ and $\zeta_0 := x_0 - x^\star$, so $\|\zeta_0\| \le D$ by assumption. Also, suppose that $n$ is a power of 2, as the adjustment on Line 2 only affects $n$ (and hence the guarantees) by constant factors. The privacy claim follows immediately from Lemma 12 assuming its preconditions are met, which we verify at the end of the proof. By applying Lemma 13 in each phase $i \in [I]$ to $x \leftarrow x_i$, assuming its preconditions are met, we have

$$
\begin{aligned}
\mathbb{E}\left[F_\mathcal{P}(x_I) - F_\mathcal{P}(x^\star)\right] &\le \sum_{i \in [I]} \left( \frac{\mathbb{E}\left[\|\zeta_{i-1}\|^2\right]}{2\eta_i n_i} + 2\eta_i G_2^2 + \frac{G_k^k D}{(\frac{C}{2})^{k-1}} \right) + \frac{G_2 DI}{n^2} \\
&\quad + \mathbb{E}\left[F_\mathcal{P}(x_k) - F_\mathcal{P}(\bar{x}_k)\right] \\
&\le \frac{4D^2}{\eta n} + \frac{\eta G_2^2}{2} + \frac{G_k^k DI}{(\frac{C}{2})^{k-1}} + \frac{G_2 D}{\sqrt{n}} + G_2 \sigma_I \sqrt{d} \\
&\quad + \sum_{i \in [I-1]} \left( \frac{3600 C^2 d\eta_i \log(\frac{3}{\delta})}{n_i \varepsilon^2} + \frac{\eta_i G_2^2}{2} \right).
\end{aligned}
$$

In the first inequality, we used $G_1 \le G_2$-Lipschitzness of $F_\mathcal{P}$ by Lemma 2, and in the second inequality, we pulled out the $i = 1$ term and adjusted indices, and bounded $I \le n$ and used Jensen's inequality to bound $(\mathbb{E}\|\zeta_I\|)^2 \le \mathbb{E}\|\zeta_I\|^2 = \sigma_I^2 d$. Now using that $\frac{\eta_i}{n_i}$ and $\eta_i$ are geometrically decaying sequences, we continue bounding the above display using our choice of $C$:

$$
\begin{aligned}
\mathbb{E}\left[F_\mathcal{P}(x_I) - F_\mathcal{P}(x^\star)\right] &\le \frac{4D^2}{\eta n} + \eta G_2^2 + \frac{14400(\frac{C}{2})^2 d\eta \log(\frac{3}{\delta})}{n\varepsilon^2} + \frac{G_k^k DI}{(\frac{C}{2})^{k-1}} + \frac{G_2 D}{\sqrt{n}} + G_2 \sigma_I \sqrt{d} \\
&\le \frac{4D^2}{\eta n} + \eta G_2^2 + 2(A\eta)^{\frac{k-1}{k+1}} \left(G_k^k DI\right)^{\frac{2}{k+1}} + \frac{G_2 D}{\sqrt{n}} + G_2 \sigma_I \sqrt{d},
\end{aligned}
$$

$$
\text{for } A := \frac{14400 d \log^2(\frac{15n}{\delta})}{n\varepsilon^2}, \; C = 2\left(\frac{G_k^k DI}{A\eta}\right)^{\frac{1}{k+1}}.
$$

Next, plugging in our choice of

$$
\eta = \min\left( \underbrace{\sqrt{\frac{4}{n}} \cdot \frac{D}{G_2}}_{:=\eta_1}, \; \underbrace{\frac{DI}{G_k n} \cdot \left(\frac{n}{A}\right)^{\frac{k-1}{2k}}}_{:=\eta_2} \right), \tag{21}
$$

we have the claimed utility bound upon simplifying, and using that $G_2 \sigma_I \sqrt{d}$ is a low-order term.

We now verify our parameters satisfy the conditions in Lemma 12 and Lemma 13, which concludes the proof. First, it is straightforward to check that both sets of conditions are implied by

$$
\frac{96\eta\beta c}{\sqrt{\varepsilon}} \log\left(\frac{30n}{\delta}\right) \le 1, \; c \ge 4n \cdot \left(\frac{2G_k}{C}\right)^k, \; \text{and } c \ge \frac{26}{\varepsilon} \log\left(\frac{15n}{\delta}\right), \tag{22}
$$

given that we chose $\omega = \frac{18}{\varepsilon}\sqrt{2c\log(\frac{15}{\delta})} \le \frac{c}{\sqrt{\varepsilon}}$. Indeed, $C \ge 8\omega_i \log(\frac{30n_i}{\delta}) \impliedby 2\eta\beta\omega \log(\frac{30n}{\delta}) \le 1$ which is subsumed by the first condition in (22). Clearly, $c \ge \frac{26}{\varepsilon} \log(\frac{15n}{\delta})$, giving the third condition in (22). Next, a direct computation with the definition of $\eta_2$ in (21) yields

$$
c = 2\sqrt{An} = 4n \cdot \sqrt{\frac{A}{n}} = 4n \cdot \left(G_k \cdot \left(\frac{A\eta_2}{G_k^k DI}\right)^{\frac{1}{k+1}}\right)^k.
$$

Now because $C$ depends inversely on $\eta \le \eta_2$ defined in (21), the second condition in (22) holds:

$$
c = 4n \cdot \left(G_k \cdot \left(\frac{A\eta_2}{G_k^k DI}\right)^{\frac{1}{k+1}}\right)^k \ge 4n \cdot \left(G_k \cdot \left(\frac{A\eta}{G_k^k DI}\right)^{\frac{1}{k+1}}\right)^k = 4n \cdot \left(\frac{2G_k}{C}\right)^k.
$$

Finally, the first condition in (22) now follows from our upper bound on $\beta$. $\qquad\square$

## D Improved Smoothness Bounds for Generalized Linear Models

In this section, we give an improved algorithm for heavy-tailed private SCO when the sample functions $f(x; s)$ are instances of a smooth generalized linear model (GLM). That is, we assume the sample space $\mathcal{S} \subseteq \mathbb{R}^d$, and that for a convex function $\sigma : \mathbb{R} \to \mathbb{R}$,

$$f(x; s) = \sigma\left(\langle s, x \rangle\right). \tag{23}$$

We also assume that all $f(x; s)$ are $\beta$-smooth. Observe that

$$\nabla f(x; s) = \sigma'(\langle s, x \rangle)s, \tag{24}$$

so that for all $x \in \mathcal{X}$, $\nabla f(x; s)$ are all scalar multiples of the same vector $s$. We prove that under this assumption, clipped gradient descent steps can only improve contraction, in contrast to Fact 16.

**Lemma 14.** *Let $s, s' \in \mathbb{R}$ and let $x, x', g \in \mathbb{R}^d$. Assume that*

$$\|(x - sg) - (x' - s'g)\| \leq \|x - x'\| .$$

*Then for any $C \geq 0$, letting $t := \text{sign}(s) \min(|s|, C)$ and $t' := \text{sign}(s') \min(|s'|, C)$, we have*

$$\|(x - tg) - (x' - t'g)\| \leq \|x - x'\| .$$

*Proof.* Note that the premise is impossible unless $\text{sign}(s - s') = \text{sign}(\langle x - x', g \rangle)$. Without loss of generality, assume they are both nonnegative, else we can negate $s, s', g$. In this case,

$$\|(x - x') - (s - s')g\| \leq \|x - x'\| \iff (s' - s)^2 \|g\|^2 \leq 2(s - s') \langle x - x', g \rangle$$

$$\iff s - s' \leq \frac{2 \langle x - x', g \rangle}{\|g\|^2}.$$

Now, observe that $t - t' \leq s - s'$ and $\text{sign}(t - t') = \text{sign}(s - s')$, for any value of $C \geq 0$. Therefore, $t - t' \leq \frac{2\langle v, g \rangle}{\|g\|^2}$ as well, and we can reverse the above chain of implications. $\qquad\square$

Note that the premise of Lemma 14 is exactly an instance of Fact 3 where $\nabla f(x)$ and $\nabla f(x')$ are scalar multiples of the same direction, which is the case for GLMs by (24). Hence, Lemma 14 shows the contraction property in Fact 3 is preserved after clipping gradients (again, for GLMs).

We can now directly combine Lemma 8 and our contraction results, used to analyze the stability of Algorithm 5, with the iterative localization framework of [FKT20], Section 4.

---

**Algorithm 8:** OnePass-Clipped-DP-SGD$(\mathcal{D}, n, \mathcal{X}, x_0, \rho)$

---

1 **Input:** Dataset $\mathcal{D} = \{s_i\}_{i \in [n]} \in \mathcal{S}^n$, domain $\mathcal{X} \subset \mathbb{B}(x_0, D)$ for $x_0 \in \mathcal{X}$

2 $I \leftarrow \lfloor \log_2(n) \rfloor$

3 $n \leftarrow 2^I$

4 $\eta \leftarrow \min(\sqrt{\frac{8}{n}} \cdot \frac{D}{G_2}, \frac{1}{n} \cdot (\frac{n^2 \rho}{32d})^{\frac{k-1}{2k}} \cdot \frac{2^{\frac{k+1}{2k}} D}{G_k}), C \leftarrow (\frac{G_k^k D \rho n}{32 \eta d})^{\frac{1}{k+1}}$

5 **for** $i \in [I]$ **do**

6 $\quad n_i \leftarrow 2^{-i}n, \eta_i \leftarrow 16^{-i}\eta, C_i \leftarrow 2^i C, \sigma_i \leftarrow 2\eta_i C_i \cdot \sqrt{\frac{2}{\rho}}$

7 $\quad \mathcal{D}_i \leftarrow$ first $n_i$ elements of $\mathcal{D}, \mathcal{D} \leftarrow \mathcal{D} \setminus \mathcal{D}_i$

8 $\quad \bar{x}_i \leftarrow$ OnePass-Clipped-SGD$(\mathcal{D}_i, C_i, \eta_i, n_i, \mathcal{X}, x_{i-1})$

9 $\quad \xi_i \sim \mathcal{N}(\mathbb{0}_d, \sigma_i^2 \mathbf{I}_d)$

10 $\quad x_i \leftarrow \bar{x}_i + \xi_i$

11 **end**

12 **Return:** $x_I$

---

**Theorem 4.** *Consider an instance of $k$-heavy-tailed private SCO, following notation in Definition 4, let $x^\star := \text{argmin}_{x \in \mathcal{X}} F_{\mathcal{P}}(x)$, and let $\rho \geq 0$. Further, assume that for a convex function $\sigma$, the sample functions $f(x; s)$ satisfy (23) for all $s \in \mathcal{S} \subseteq \mathbb{R}^d$. Finally, assume $f(x; s)$ is $\beta$-smooth for all $s \in \mathcal{S}$,*

where $\beta \leq \max(\sqrt{\frac{n}{2}} \cdot \frac{G_2}{D}, n \cdot (\frac{d}{n^2 \rho})^{\frac{k-1}{2k}} \cdot \frac{G_k}{D})$. Algorithm 8 is a $\rho$-CDP algorithm which draws $\mathcal{D} \sim \mathcal{P}^n$, queries $n$ sample gradients (using samples in $\mathcal{D}$), and outputs $x_I \in \mathcal{X}$ satisfying

$$\mathbb{E}\left[F_{\mathcal{P}}(x_I) - F_{\mathcal{P}}(x^\star)\right] \leq 4G_2 D \sqrt{\frac{1}{n}} + 26 G_k D \left(\frac{\sqrt{d}}{n\sqrt{\rho}}\right)^{1-\frac{1}{k}}.$$

*Proof.* We begin with the privacy claim. Consider neighboring datasets $\mathcal{D}$, $\mathcal{D}'$, and suppose the datasets differ on the $j^{\text{th}}$ entry such that $s_j \in \mathcal{D}_i$ (if the differing entry is not in $\cup_{i \in [I]} \mathcal{D}_i$, Algorithm 8 clearly satisfies 0-CDP). Let $\bar{x}_i$ and $\bar{x}_i'$ be the outputs of Line 8 when run with the same initialization $x_{i-1}$, and neighboring $\mathcal{D}_i$, $\mathcal{D}_i'$. By the assumption on $\beta$, since $\eta_i \leq \eta$ for all $i \in [I]$, we can apply Fact 3 and Lemma 14 (recalling the characterization (24)) to show $\|\bar{x}_i - \bar{x}_i'\| \leq 2\eta_i C_i$ with probability 1. Therefore, by our choice of $\sigma_i$ and the first and third parts of Lemma 1, the whole algorithm is $\rho$-CDP regardless of which $\mathcal{D}_i$ contained the differing sample, since all other calls to OnePass-Clipped-SGD are 0-CDP as we can couple all randomness used by the calls.

Next, we prove the utility claim. For simplicity, let $\bar{x}_0 := x^\star$ and $\xi_0 := x_0 - x^\star$, so $\|\xi_0\| \leq D$ by assumption. By applying Lemma 8 for all $i \in [I]$ with $x_0 \leftarrow x_{i-1}$ and $u \leftarrow \bar{x}_{i-1}$, we have

$$\mathbb{E}\left[F_{\mathcal{P}}(x_I) - F_{\mathcal{P}}(x^\star)\right] = \sum_{i \in [I]} \mathbb{E}\left[F_{\mathcal{P}}(\bar{x}_i) - F_{\mathcal{P}}(\bar{x}_{i-1})\right] + \mathbb{E}\left[F_{\mathcal{P}}(x_I) - F_{\mathcal{P}}(\bar{x}_I)\right]$$

$$\leq \sum_{i \in [I]} \left(\mathbb{E}\left[\frac{\|\xi_{i-1}\|^2}{2\eta_i n_i}\right] + \frac{\eta_i G_2^2}{2} + \frac{G_k^k D}{(k-1)C_i^{k-1}}\right) + G_1 \mathbb{E}\left[\|x_I - \bar{x}_I\|\right]$$

$$\leq \frac{4D^2}{\eta n} + \sum_{i \in [I-1]} 2^{-i} \left(\frac{32 d\eta C^2}{\rho n} + \frac{\eta G_2^2}{2} + \frac{G_k^k D}{C^{k-1}}\right) + \sqrt{\frac{8d}{\rho}} G_1 \eta C \cdot 8^{-I}$$

$$\leq \frac{4D^2}{\eta n} + \frac{32 d\eta C^2}{\rho n} + \frac{\eta G_2^2}{2} + \frac{G_k^k D}{C^{k-1}} + 24\sqrt{\frac{d}{\rho}} \cdot \frac{G_1 \eta C}{n^3},$$

where the second line applied Lemma 2, the third used Jensen's inequality to bound $\mathbb{E}[\|x_I - \bar{x}_I\|]^2 \leq \mathbb{E}[\|x_I - \bar{x}_I\|^2]$ and our assumption $k \geq 2$, and the last used the geometric decay of the different parameters. Finally, by plugging in our choices of $C, \eta$, we have

$$\frac{4D^2}{\eta n} + \frac{\eta G_2^2}{2} + \frac{32 d\eta C^2}{\rho n} + \frac{G_k^k D}{C^{k-1}} = \frac{4D^2}{\eta n} + \frac{\eta G_2^2}{2} + 2\eta^{\frac{k-1}{k+1}} \left(G_k^k D\right)^{\frac{2}{k+1}} \left(\frac{32d}{\rho n}\right)^{\frac{k-1}{k+1}}$$

$$\leq G_2 D \sqrt{\frac{8}{n}} + 8 G_k D \left(\frac{\sqrt{d}}{n\sqrt{\rho}}\right)^{1-\frac{1}{k}}.$$

We can also check that the final summand is a low-order term, by using $\eta \leq \frac{1}{n} \cdot \left(\frac{n^2 \rho}{32d}\right)^{\frac{k-1}{2k}} \cdot \frac{2^{\frac{k+1}{2k}} D}{G_k}$:

$$24\sqrt{\frac{d}{\rho}} \cdot \frac{G_1 \eta C}{n^3} \leq \frac{5 G_k D}{n^2}.$$

The conclusion follows by adjusting $n$, since Algorithm 8 is run with a sample count in $\left[\frac{n}{2}, n\right]$. $\square$

# E High-probability stochastic convex optimization

In this section, to highlight another application of our population-level localization framework, we show that it obtains improved high-probability guarantees for the following standard bounded-variance estimator parameterization of SCO in the non-private setting.

**Definition 6** (Stochastic convex optimization). *Let $\mathcal{X} \subset \mathbb{R}^d$ be compact and convex, with $\text{diam}(\mathcal{X}) = D$. In the* stochastic convex optimization (SCO) *problem, there is a convex function $f : \mathcal{X} \to \mathbb{R}$, and we have query access to a stochastic oracle $g : \mathcal{X} \to \mathbb{R}^d$ satisfying, for all $x \in \mathcal{X}$,*

$$\mathbb{E}\left[g(x)\right] \in \partial f(x), \ \mathbb{E}\left[\|g(x)\|^2\right] \leq G^2.$$

*For a convex function $\psi : \mathcal{X} \to \mathbb{R}$, our goal in SCO is to optimize the composite function $f + \psi$.*

For instance, one can set $\psi$ to the constant zero function to recover the non-composite variant of SCO. We include the composite variant of Definition 6 as it is a standard extension in the SCO literature, under the assumption that the function $\psi$ is "simple." The specific notion of simplicity we use is that $\psi : \mathcal{X} \to \mathbb{R}$ admits an efficient *proximal oracle* (Definition 7).

**Definition 7** (Proximal oracle). *Let $\mathcal{X} \subset \mathbb{R}^d$ be compact and convex. We say $\mathcal{O}$ is a* proximal oracle *for a convex function $\psi : \mathcal{X} \to \mathbb{R}$ if for any inputs $v \in \mathbb{R}^d$, $\eta \in \mathbb{R}_{\geq 0}$, $\mathcal{O}(v)$ returns*

$$\operatorname*{argmin}_{x \in \mathcal{X}} \left\{ \frac{1}{2\eta} \|x - v\|^2 + \psi(x) \right\}.$$

In Theorem 5, we give an algorithm which uses $n$ queries to each of $g$ and a proximal oracle for $\psi$, and achieves an error bound for $f + \psi$ of

$$O\left( GD \cdot \sqrt{\frac{\log \frac{1}{\delta}}{n}} \right), \tag{25}$$

with probability $\geq 1 - \delta$. Similar rates are straightforward to derive using martingale concentration when the estimator $g$ is assumed to satisfy heavier tail bounds, such as a sub-Gaussian norm. To our knowledge, the rate (25) was first attained recently by [CH24], who also proved a matching lower bound. Our Theorem 5 gives an alternative route to achieving this error bound. As was the case in several recent works in the literature [HS16, DDXZ21, Lia24] who studied high-probability variants of stochastic convex optimization, our Theorem 5 is based on using geometric aggregation techniques within a proximal point method framework (in our case, using Fact 2 within Algorithm 2). However, these aforementioned prior works all assume additional smoothness bounds on the function $f$.

We use the following standard result in the literature as a key subroutine.

**Lemma 15** (Lemma 1, [ACJ$^+$21]). *In the setting of Definition 6, assume $\psi$ is $\lambda$-strongly convex, let $x^\star := \operatorname{argmin}_{x \in \mathcal{X}} f(x) + \psi(x)$, and let $T \in \mathbb{N}$. There is an algorithm which queries the stochastic oracle $g$ and a proximal oracle for $\psi$ each $T$ times, and produces $\bar{x}$ satisfying, with probability $\geq \frac{4}{5}$,*

$$\|\bar{x} - x^\star\| \leq \frac{30G}{\lambda \sqrt{T}}.$$

We combine Lemma 15 with Proposition 2 to obtain the following high-probability SCO algorithm.

**Theorem 5.** *Consider an instance of SCO, following notation in Definition 6, let $n \in \mathbb{N}$, $x^\star := \operatorname{argmin}_{x \in \mathcal{X}} f(x) + \psi(x)$, and $\delta \in (0, \frac{1}{2})$. There is an algorithm using $n$ queries to $g$ and a proximal oracle for $\psi$ and outputs $x \in \mathcal{X}$ satisfying, for a universal constant $C_{\mathrm{sco}}$, with probability $\geq 1 - \delta$,*

$$f(x) + \psi(x) - f(x^\star) - \psi(x^\star) \leq C_{\mathrm{sco}} \cdot GD \cdot \sqrt{\frac{\log \frac{1}{\delta}}{n}}.$$

*Proof.* Assume without loss of generality that $\frac{1}{\delta}$ is a sufficiently large constant (else we can adjust the constant factor $C_{\mathrm{sco}}$), and that $n$ is sufficiently larger than $\log \frac{1}{\delta}$ (else the result holds because the range of the function is bounded by $GD$). We instantiate Proposition 2 with $F_{\mathcal{P}} \leftarrow f + \psi$, $I \leftarrow \frac{1}{2} \log_2 n$, and in each phase $i \in [I]$ of Algorithm 2, we let $n_i := \frac{n}{2^i}$. In the remainder of the proof, we describe how to implement (8) in the $i^{\text{th}}$ phase, where $F_{\mathcal{P}} \leftarrow f + \psi$, splitting into cases.

If $\frac{1}{\delta}$ is bounded by polylog$(n)$ and $n$ is sufficiently large, suppose that $n$ is a power of 4, else we can use fewer queries and lose a constant factor in the guarantee. Then we can use a batch of $n_i$ consecutive queries, divided into $48 \log(\frac{1}{\delta_i})$ portions, where $\delta_i := \frac{\delta}{2^i}$. We then use Lemma 15 on each portion of queries, with $f \leftarrow f$ and $\psi \leftarrow \psi + \frac{\lambda_i}{2} \|\cdot - x_{i-1}\|^2$; it is straightforward to see that Definition 7 generalizes to give a proximal oracle for this new $\psi$. A Chernoff bound shows that at least $\frac{3}{5}$ of the portions will return a point satisfying the bound in Lemma 15 except with probability $\delta_i$, so Fact 2 returns us a point at distance at most $\frac{90G}{\lambda \sqrt{T}}$ from $x_i^\star$, where

$$T = \Omega\left( \frac{n_i}{\log \frac{1}{\delta_i}} \right) = \Omega\left( \frac{n}{2^i \left( \log \frac{1}{\delta} + i \right)} \right),$$

(accounting for rounding error). Therefore, (8) holds with

$$\Delta = \frac{C_{\mathrm{sco}}}{2} \cdot G \cdot \sqrt{\frac{\log \frac{1}{\delta}}{n}},$$

for sufficiently large $C_{\mathrm{sco}}$. Proposition 2 then implies that Algorithm 2 outputs $x$ satisfying

$$f(x) + \psi(x) - f(x^\star) - \psi(x^\star) \leq 2GD \cdot \sqrt{\frac{\Delta}{n^{1.5}}} + \frac{C_{\mathrm{sco}}}{2} \cdot GD \cdot \sqrt{\frac{\log \frac{1}{\delta}}{n}} \leq C_{\mathrm{sco}} \cdot GD \cdot \sqrt{\frac{\log \frac{1}{\delta}}{n}},$$

where we use that $G_1 \leq G$ by Jensen's inequality and our second moment bound in Definition 6. The failure probability follows from a union bound because we ensured that $\sum_{i \in [I]} \delta_i \leq \delta$.

Finally, if $\frac{1}{\delta}$ is larger than $\mathrm{polylog}(n)$, then we let $I, J \in \mathbb{N}$ be chosen such that

$$I := \left\lfloor \log_2 \left( \frac{n}{J} \right) \right\rfloor, \quad J \geq 48 \log \left( \frac{I}{\delta} \right),$$

which is achievable with $I = O(\log n)$ and $J = O(\log \frac{\log n}{\delta}) = O(\log \frac{1}{\delta})$. Let $m := \frac{n}{J}$, and assume without loss that $m$ is a power of 2, which we can guarantee by discarding $\leq \frac{1}{2}$ our queries, losing a constant factor in the error bound. The remainder of the proof follows identically to the first part of this proof, where we union bound over $I$ phases, the $i^{\mathrm{th}}$ of which uses $J$ batches of $\frac{m}{2^i}$ unused queries. Again we may apply Lemma 15 and Fact 2 with $T = \frac{m}{2^i}$, so (8) holds with

$$\Delta = \frac{C_{\mathrm{sco}}}{2} \cdot G \cdot \sqrt{\frac{\log \frac{1}{\delta}}{n}},$$

except with probability $\frac{\delta}{I}$. The conclusion then follows from Proposition 2. $\qquad\square$

## F   Non-contraction of truncated contractive steps

In this section, we demonstrate that a natural conjecture related to the performance of clipped private gradient algorithms in the smooth setting is false. We state this below as Conjecture 1. To motivate it, suppose $v$ is the difference between a current pair of coupled iterates of a private gradient algorithm instantiated on neighboring datasets, and suppose the differing sample function has already been encountered. If we take a coupled gradient step in a sufficiently smooth function, Fact 3 shows that the step is a contraction. However, to preserve privacy in the heavy-tailed setting, it is natural to ask whether such a contractive step remains contractive after the gradients are clipped, i.e. the statement of Conjecture 1 (which gives the freedom for $C$ to be lower bounded).

**Conjecture 1.** *Let $\|v\|_2 \leq C$ for a sufficiently large constant $C$, and let $\|v - (g - h)\| \leq \|v\|$. Let $g' = \Pi_1(g)$ and $h' = \Pi_1(h)$.[9] Then, $\|v - (g' - h')\| \leq C$.*

We strongly refute Conjecture 1, by disproving it for any $C \geq 0$. We remark that Lemma 16 does not necessarily rule out this approach to designing heavy-tailed DP-SCO algorithms in the smooth regime, but demonstrates an obstacle if additional structure of gradients is not exploited.

**Lemma 16.** *Conjecture 1 is false for any choice of $C \geq 0$.*

*Proof.* We give a 2-dimensional counterexample. Let

$$v = \begin{pmatrix} -C \\ 0 \end{pmatrix}, \quad g = \begin{pmatrix} 1 \\ 0 \end{pmatrix}, \quad h = \begin{pmatrix} \frac{2C+1}{C+1} \\ \frac{C\sqrt{2C+1}}{C+1} \end{pmatrix} = \sqrt{2C+1} \underbrace{\begin{pmatrix} \frac{\sqrt{2C+1}}{C+1} \\ \frac{C}{C+1} \end{pmatrix}}_{:=h'}.$$

Observe that

$$v - (g - h) = \begin{pmatrix} -(C+1) + \frac{2C+1}{C+1} \\ \frac{C\sqrt{2C+1}}{C+1} \end{pmatrix} = \begin{pmatrix} \frac{-C^2}{C+1} \\ \frac{C\sqrt{2C+1}}{C+1} \end{pmatrix} = C \begin{pmatrix} \frac{-C}{C+1} \\ \frac{\sqrt{2C+1}}{C+1} \end{pmatrix}.$$

---

[9] By scale-invariance of the claim, the assumption that the truncation threshold is 1 is without loss of generality.

It is easy to verify $\|v - (g - h)\| = C$ at this point. Moreover,

$$v - (g' - h') = \begin{pmatrix} -(C+1) + \frac{\sqrt{2C+1}}{C+1} \\ \frac{C}{C+1} \end{pmatrix}.$$

For $C \geq 0$, the first coordinate of this vector is already less than $-C$. $\qquad\square$

## G   Non-decay of empirical squared bias

In this section, we present an obstacle towards a natural approach to improving the logarithmic terms in our algorithm in Section 3. We follow the notation of Section 3.1, i.e. for samples $\{i \equiv s_i\}_{i \in [n]} \sim \mathcal{P}^n$, we define sample functions $f_i \equiv f(\cdot; s_i)$, and let

$$b_{\mathcal{D}} := \max_{x \in \mathcal{X}} \left\| \frac{1}{n} \sum_{i \in [n]} \nabla f_i(x) - \frac{1}{n} \sum_{i \in [n]} \Pi_C(\nabla f_i(x)) \right\|. \tag{26}$$

A basic bottleneck with known approaches following SCO-to-ERM reductions is that they require a strongly convex ERM solver as a primitive, due to known barriers to generalization in SCO without strong convexity (see e.g. discussion in [SSSSS09]). This poses an issue in the heavy-tailed setting, because standard analyses of strongly convex clipped SGD (see e.g. our Proposition 1) appear to suffer a dependence on $b_{\mathcal{D}}^2$ in the utility bound, which upon taking expectations requires bounding

$$\mathbb{E}_{\mathcal{D} \sim \mathcal{P}^n} b_{\mathcal{D}}^2 = \mathbb{E}_{\mathcal{D} \sim \mathcal{P}^n} \left[ \max_{x \in \mathcal{X}} \left\| \frac{1}{n} \sum_{i \in [n]} \nabla f_i(x) - \frac{1}{n} \sum_{i \in [n]} \Pi_C(\nabla f_i(x)) \right\|^2 \right]. \tag{27}$$

Recall from Lemma 3 that it is straightforward to bound $\mathbb{E} b_{\mathcal{D}} \leq \frac{G_k^k}{C^{k-1}}$, due to Fact 1. Bounding $\mathbb{E} b_{\mathcal{D}}^2$ is more problematic; in [LR23], requiring this bound resulted in a dependence on $G_{2k}$ as opposed to $G_k$ (see the proof of Theorem 31), which we avoid (up to a polylogarithmic overhead) via our population-level localization strategy. We now present an alternative strategy to bound (27), avoiding a $G_{2k}$ dependence. Observe that, by using $(a + b + c)^2 \leq 3(a^2 + b^2 + c^2)$,

$$\mathbb{E}_{\mathcal{D} \sim \mathcal{P}^n} b_{\mathcal{D}}^2 \leq 3 \underbrace{\mathbb{E}_{\mathcal{D} \sim \mathcal{P}^n} \left[ \max_{x \in \mathcal{X}} \left\| \frac{1}{n} \sum_{i \in [n]} \nabla f_i(x) - \nabla F_{\mathcal{P}}(x) \right\|^2 \right]}_{:=T_1}$$

$$+ 3 \underbrace{\max_{x \in \mathcal{X}} \left\| \nabla F_{\mathcal{P}}(x) - \mathbb{E}_{s \sim \mathcal{P}} \left[ \Pi_C(\nabla f(x; s)) \right] \right\|^2}_{:=T_2} \tag{28}$$

$$+ 3 \underbrace{\mathbb{E}_{\mathcal{D} \sim \mathcal{P}^n} \left[ \max_{x \in \mathcal{X}} \left\| \frac{1}{n} \sum_{i \in [n]} \Pi_C(\nabla f_i(x)) - \mathbb{E}_{s \sim \mathcal{P}} [\Pi_C(\nabla f(x; s))] \right\|^2 \right]}_{:=T_3}.$$

We focus on $T_1$, as $T_3$ can be bounded by similar means (as truncation can only improve moment bounds), and $T_2 \leq \frac{G_k^{2k}}{C^{2(k-1)}}$ via Fact 1. Hence, if we can show that $T_1 = O(\frac{G_2^2}{n})$ under the moment bound assumption in Assumption 1, we can avoid the logarithmic factors lost by our population localization approach. We suggest the following conjecture as an abstraction of this bound.

**Conjecture 2.** *Let $\mathcal{P}$ be a distribution over $\mathcal{S}$. For each $x \in \mathcal{X}$, let $g(x; s) \in \mathbb{R}^d$ be a random vector, indexed by $s \sim \mathcal{S}$, satisfying $\mathbb{E}_{s \sim \mathcal{P}}[g(x; s)] = \mathbb{0}_d$ and $\mathbb{E}_{s \sim \mathcal{P}}[\sup_{x \in \mathcal{X}} \|g(x; s)\|^2] \leq 1$. Finally for $S \sim \mathcal{P}^n$ and $x \in \mathcal{X}$, let $g(x; S) := \frac{1}{n} \sum_{s \in S} g(x; s)$. Then,*

$$\mathbb{E}_{S \sim \mathcal{P}^n} \left[ \sup_{x \in \mathcal{X}} g(x; S)^2 \right] = O\left( \frac{1}{n} \right).$$

Note that the bound in Conjecture 2 exactly corresponds to $T_1$ in (28), after rescaling all sample gradients by $\frac{1}{G_2}$, and centering them by subtracting $\nabla F_{\mathcal{P}}(x)$. Hence, if Conjecture 2 is true, it would yield the following desirable bound in (28):

$$\mathbb{E}_{\mathcal{D}\sim\mathcal{P}^n} b_{\mathcal{D}}^2 = O\left(\frac{G_2^2}{n} + \frac{G_k^k}{(k-1)C^{k-1}}\right).$$

Moreover, it is simple to prove a bound of $O(1)$ on the right-hand side of Conjecture 2, and as $n \to \infty$ it is reasonable to suppose $g(x; S) \to \mathbb{0}_d$ for all $x \in \mathcal{X}$. Nonetheless, we refute Conjecture 2 in full generality with a simple 1-dimensional example.

**Lemma 17.** *Conjecture 2 is false.*

*Proof.* Let $\mathcal{S} = [0, 1]$ and let $\mathcal{P}$ be the uniform distribution over $\mathcal{S}$. Let $\mathcal{X}$ index a set of random $g(x; \cdot) : [0, 1] \to [0, 1]$ which are nonzero at finitely many points.[10] Then $\mathbb{E}_{s\sim\mathcal{P}} g(x; s) = 0$ for all $x \in \mathcal{X}$, and $g(x; s)^2 \le 1$ for all $x \in \mathcal{X}, s \in \mathcal{S}$. However, for any $S \in [0, 1]^n$, we have

$$\sup_{x\in\mathcal{X}} g(x; S)^2 = 1.$$

$\square$

While Lemma 17 does not rule out the approach suggested in (28) (or other approaches) to improve the analysis of strongly convex ERM solvers in heavy-tailed settings, it presents an obstacle to applying the natural decomposition strategy in (28). To overcome Lemma 17, one must either use more structure about the index set $\mathcal{X}$ or the iterates encountered by the algorithm, or consider a different decomposition strategy for bounding the squared empirical bias.

## H   Proof of Lemma 11

In this section, we prove Lemma 11. We first require the following standard fact (see e.g. [Sch14]).

**Fact 4.** *Let $\mathcal{X} \subseteq \mathbb{R}^d$ be a convex set. Then for any $x, y \in \mathbb{R}^d$, we have*

$$\|\Pi_{\mathcal{K}}(x) - \Pi_{\mathcal{K}}(y)\| \le \|x - y\|.$$

We now set up some notation. Let $\{\psi_j : \mathcal{X} \to \mathcal{X}\}_{j\in[T]}$ and $\{\phi_j : \mathcal{X} \to \mathcal{X}\}_{j\in[T]}$ be two sequences of operations. We say that an operation pair $(\psi, \phi)$ is contractive if for any two points $x, y \in \mathcal{X}$,

$$\|\psi(x) - \phi(y)\| \le \|x - y\|.$$

We say an operation pair $(\psi, \phi)$ is $(C, \zeta)$-contractive if for any $x, y$ where $\|x - y\| \le C$, we have

$$\|\psi(x) - \phi(y)\| \le \|x - y\| + \zeta.$$

Let $\psi^j(x) = \psi_j \circ \psi_{j-1} \circ \ldots \circ \psi_1(x)$, and define $\phi^j$ similarly, for all $j \in [T]$.

We prove Lemma 11 as a consequence of the following more general result.

**Lemma 18.** *Let $x_0 = x_0' \in \mathcal{X}$, and consider two sequences of operations $\{\psi_j : \mathcal{X} \to \mathcal{X}\}_{j\in[T]}$ and $\{\psi_j' : \mathcal{X} \to \mathcal{X}\}_{j\in[T]}$ satisfying the following conditions, for $c := \lfloor \frac{C}{\zeta} \rfloor$.*

*1. For at least $T - c - 1$ indices $j \in [T]$, $(\psi_j, \phi_j)$ is contractive.*

*2. At most one operation pair, $(\psi_k, \psi_k)$, is $(\infty, C)$-contractive.*

*3. For at most $c$ indices $j \in [T]$, $(\psi_j, \phi_j)$ is $(2C, \zeta)$-contractive.*

*Then for all $j \in [T]$, we have that $\|\psi^j(x_0) - \phi^j(y_0)\| \le 2C$.*

---

[10]We note there is a bijection between $\mathcal{X}$ and any convex subset $\mathcal{X}'$ of $\mathbb{R}^d$ containing a ball with nonzero radius. To see this, it is well-known that there is a bijection from $[0, 1]$ to $\mathbb{R}_{\ge 0}$, and we can simply construct a bijection between $\mathbb{R}_{\ge 0}$ and $\mathcal{X}$ by mapping the interval $[i - 1, i]$ to $[0, 1]^{2i}$ (where the first $i$ coordinates specify the nonzero points, and the next $i$ coordinates specify their values) for all $i \in \mathbb{N}$. Finally, it is well-known there is a bijection between $[0, 1]$ and $\mathbb{R}^d$, and we can construct a bijection between $\mathcal{X}'$ and $\mathbb{R}^d$ by considering each 1-dimensional projection separately.

*Proof.* Define $\Delta_j := \|\psi^j(x_0) - \phi^j(x'_0)\|$ for all $j \in [T]$. Let $a_j \leq c$ be the total number of $(2C, \zeta)$-contractive operation pairs $(\psi_i, \phi_i)$ where $i \leq j$, and let $b_j$ be the 0-1 indicator variable for $k \leq j$. We use induction to show that $\Delta_j \leq a_j\zeta + b_jC$. When $j = 1$, the claim holds. Now if the claim holds for $j - 1$, then $\Delta_{j-1} \leq a_{j-1}\zeta + b_{j-1}C \leq 2C$. Hence, by definition,

$$\Delta_j \leq \Delta_{j-1} + (a_j - a_{j-1})\zeta + (b_j - b_{j-1})C = a_j\zeta + b_jC,$$

which completes our induction. This also implies $\Delta_T \leq 2C$ as claimed. $\qquad\square$

*Proof of Lemma 11.* Throughout the following proof, note that $\hat{c}_i \leq 2c$ deterministically (due to our use of $\mathrm{BLap}(\frac{3}{\varepsilon}, c)$ noise), and under the stated parameter bounds,

$$\hat{C} \in \left[\frac{7C}{8}, \frac{9C}{8}\right] \text{ and } |\nu_{i,j}| \leq \frac{C}{4} \text{ for all } j \in [n_i].$$

Let $\{g_{i,j} = \Pi_C(\nabla f(x_{i,j}; s_{i,j}))\}_{j \in [n_i]}$ and $\{g'_{i,j} = \Pi_C(\nabla f(x'_{i,j}; s'_{i,j}))\}$ be the two truncated gradient sequences in the $i^{\text{th}}$ phase corresponding to the two datasets, and let $\{x_{i,j}\}_{j \in [n_i]}$ and $\{x'_{i,j}\}_{j \in [n_i]}$ be the corresponding iterate sequences. We set the operation sequences $\psi_j(x) := \Pi_{\mathcal{X}}(x - \eta_i g_{i,j})$ and $\phi_j(x) := \Pi_{\mathcal{X}}(x - \eta_i g'_{i,j})$. We bound the contractivity of these operation pairs and apply Lemma 18.

First, note that because $\mathrm{count}_t$, $\mathrm{count}'_t < \hat{c}_i \leq 2c$, the operation pair $(\psi_j, \phi_j)$ is an identical untruncated gradient mapping for at least $t - 2c - 1$ indices $j \in [t]$. Because we assume each sample function $f(\cdot; s)$ is $\beta$-smooth, it follows that for these indices $j \in [t]$, the operation pair $(\psi_j, \phi_j)$ is contractive, by applying Fact 3, Fact 4, and $\eta_i\beta \leq 1$.

Next, recall the assumption that the datasets $\mathcal{D}, \mathcal{D}'$ differ in the $j_0^{\text{th}}$ sample only. Because $\|g_{i,j_0}\| \leq \frac{9C}{8} + \frac{C}{4} \leq \frac{11C}{8}$ by assumption, and similarly $\|g'_{i,j_0}\| \leq \frac{11C}{8}$, it follows that the operation pair $(\psi_{j_0}, \phi_{j_0})$ is $(\infty, 3C\eta_i)$-contractive by applying the triangle inequality and Fact 4.

For all remaining indices $j \in [t]$, $\mathrm{count}_t$ and $\mathrm{count}'_t$ both incremented (under the assumption that $Y_{i,j} = Y'_{i,j}$ for these indices). We claim that $(\psi_j, \phi_j)$ is $(6\eta_i C, 12\eta_i^2 C\beta)$-contractive for these iterations. To see this, we bound

$$\begin{aligned}
\|\psi_j(x_{i,j}) - \phi_j(x'_{i,j})\| &\leq \|(x_{i,j} - \eta_i g_{i,j}) - (x'_{i,j} - \eta_i g'_{i,j})\| \\
&\leq \|(x_{i,j} - \eta_i \nabla f(x_{i,j}; s_{i,j})) - (x'_{i,j} - \eta_i \nabla f(x'_{i,j}; s_{i,j}))\| \\
&\quad + \eta_i \|\nabla f(x_{i,j}; s_{i,j}) - \nabla f(x'_{i,j}; s_{i,j})\| + \eta_i \|g_{i,j} - g'_{i,j}\| \\
&\leq \|x_{i,j} - x'_{i,j}\| + 12\eta_i^2 C\beta.
\end{aligned}$$

The first line used Fact 4, the second used the triangle inequality, and the last used Fact 3, Fact 4, and the fact that $\|\nabla f(x_{i,j}; s_{i,j}) - \nabla f(x'_{i,j}; s_{i,j})\| \leq 6\eta_i C\beta$ by smoothness, when $\|x_{i,j} - x'_{i,j}\| \leq 6C\eta_i$.

Finally, it suffices to apply Lemma 18 with $C \leftarrow 3C\eta_i$, $\zeta \leftarrow 12\eta_i^2 C\beta$, and $c \leftarrow 2c$, which we can check meets the conditions of Lemma 18 under the stated parameter bounds. $\qquad\square$

