# OpenReview forum: "Private Stochastic Convex Optimization with Heavy Tails: Near-Optimality from Simple Reductions"
_NeurIPS.cc/2024/Conference — NeurIPS 2024 poster_

### Official Review · Reviewer_1QCz · 2024-07-12

**Soundness:** 3
**Presentation:** 1
**Contribution:** 2
**Rating:** 5
**Confidence:** 3

**Summary:**

The paper investigates the problem of differentially private stochastic convex optimization (SCO) under the heavy-tailed setting and achieves a nearly optimal rate of $G_2 \cdot \frac{1}{\sqrt{n}}+G_k\left(\frac{\sqrt{d}}{n \varepsilon}\right)^{1-\frac{1}{k}}$. Specifically, it first provides results using Clipped-DP-SGD in the differentially private empirical risk minimization (DP-ERM) framework and then utilizes generalization techniques to offer similar results in the population case. Finally, it explores the heavy-tailed DPSCO.

**Strengths:**

1. The paper provides a nearly optimal bound for DPSCO in the heavy-tailed setting.

**Weaknesses:**

1. Section 3 is not very clear. The connection between Sections 3.1 and 3.2 is not well explained.
2. The presentation needs improvement. One of the most important parts, Population-level Localization, is placed on the last page and is only briefly introduced and discussed. For example, how to choose parameters like $\lambda$ in Algorithm 2 and what $\Delta 4^i$ in Equation 8 represents should be explained.

**Questions:**

NA

---

> ### Author Rebuttal · Authors · 2024-08-06
>
> Thank you for your reviewing efforts and feedback.
>
> Section 3: We apologize for the lack of clarity in Section 3. In Section 3.1, we propose Algorithm 1, which achieves good performance on an empirical loss assuming the dataset satisfies a property (bounded $b_{\mathcal{D}}$, see (6)). In Section 3.2, we show we can use Algorithm 1 to get a solution with respect to the population function if the dataset is drawn from a heavy-tailed distribution. We included a description of this relationship at the start of Section 3 (Lines 226-235), but will include more connective tissue between Sections 3.1 and 3.2.
>
> Presentation: Due to page limitations, we did not add many details about the population-level localization in the main paper (some details were deferred to the appendix). We will revise our final version and add more details and intuitions to clarify it. Currently, a short description is given in Lines 85-92, and full proof of its guarantees is given in Proposition 2 in the main body. We provided additional intuition of why we developed this technical tool and its comparison to prior work in our response to Reviewer KReZ, and will incorporate this response into our revision. For your other questions, $\lambda$ is a hyperparameter that we optimize in Line 295 to minimize the expression on the previous Line 294, and $\Delta$ is a scaling of the objective error defined in Line 301.
>
> We hope this discussion was clarifying, addressed some of your concerns, and elevated the merits of our paper in your view.

---

> > ### Comment · Reviewer_1QCz · 2024-08-08
> > **Response to authors**
> >
> > Thanks for the clarification. I look forward to seeing more discussion in the revised paper. I've increased my score.

---

### Official Review · Reviewer_KReZ · 2024-07-16

**Soundness:** 3
**Presentation:** 3
**Contribution:** 3
**Rating:** 6
**Confidence:** 2

**Summary:**

This paper addresses differentially private stochastic convex optimization (DP-SCO) with heavy-tailed gradients, where previous assumptions of uniform Lipschitz constants are relaxed to bounded k-th moments. The authors introduce a new reduction-based framework that adapts strategies from the uniform Lipschitz setting, enabling optimal rates up to logarithmic factors under (ε,δ)-approximate differential privacy. They propose several algorithms, including an optimal algorithm for known Lipschitz constants, a near-linear time algorithm for smooth functions, and an optimal linear-time algorithm for smooth generalized linear models. A novel population-level localization framework is also presented, overcoming technical barriers and providing robust bounds on excess population loss without stringent gradient moment assumptions. This work advances the theoretical and practical understanding of DP-SCO with heavy-tailed gradients, outperforming previous approaches in handling real-world data challenges.

**Strengths:**

1. The paper introduces a novel reduction-based framework for handling heavy-tailed gradients in differentially private stochastic convex optimization (DP-SCO). This innovative approach enables the achievement of near-optimal rates and overcomes limitations of previous methods, marking a significant advancement in the field.
2. The proposed method bypasses the need for bounding $\mathbb{E}b_{\mathcal{D}}^2$

**Weaknesses:**

1. Referring to Corollary 2 in Line 4 of Algorithm 3 reduces readability. To enhance clarity, it would be beneficial to specify the algorithm $\mathcal{A}$ (from Corollary 2) clearly, at least within the proof. While leaving the optimization oracle as a black box might help generalize the framework, it makes the algorithm harder to follow if the instantiations are not highlighted.
2. The motivation of the population-level localization could be further clarified.

**Questions:**

1. To solve population-level localization, could one use the same (or slightly adjusted) method for the localization of the empirical minimizer?
2. How does the sample split parameter $J$ affect the utility of Algorithm 3?
3. Should $G_2^2$ and $G_k^k$ on line 7 be $G_2$ and $G_k$?

---

> ### Author Rebuttal · Authors · 2024-08-06
>
> Thank you for your careful reading; we address your questions here.
>
> Re: clarity, we agree with your suggestion, and will add a description of the algorithm and the specific guarantees we are using about it to help the reader. Thanks for pointing this out.
>
> Re: population-level localization, we will add an additional description after Line 85. The motivation for population-level localization is that we wish to aggregate empirical solutions to multiple datasets, some of which satisfy assumptions of Corollary 1 (i.e., have small $b_{\mathcal{D}}$), and some of which do not. However, each dataset has a different empirical minimizer, so it is unclear which to argue about convergence to. We instead aggregate solutions close to a population-level minimizer, shared across datasets. Previous localization frameworks could not handle the setting we consider, as described in Section 1.2, prompting our new development. In particular, previous localization frameworks used non-strongly convex losses, which have known difficulties in generalizing to population-level objectives [SSSS09].
>
> Q1: Our population-level localization is inspired by and closely related to frameworks used to minimize an empirical loss. Both rely on stochastic access to the input, but a key difference is that we cannot verify our empirical dataset actually satisfies a key assumption used to get the optimal rate (bounded $b_{\mathcal{D}}$), so the empirical solution could be meaningless. This is why we target population-level objectives, which can aggregate multiple datasets.
>
> Q2: Suppose that we can get $x_{i,j}$ such that $\|x_{i,j}-x_i\|\le R(i,J)||$ with probability at least 0.55. Then $J$ can not be too small, to make use of a Chernoff bound to show at least half of the $x_{i,j},j\in [J]$ are close to $x_i$. On the other hand, the larger $J$ is, the smaller the privacy budget we can use to generate $x_{i,j}$, since we need to compose over $J$ instances, which hurts our final utility bound. Hence $J$ can not be too small, or too large.
>
> Q3: We believe Line 7 is correct as stated. As assumed in Line 67, $G_j^j$ is the $j^{\text{th}}$ moment.
>
> We appreciate you found our approach innovative, and that our result was a significant advancement. We hope our response was clarifying, and that it elevates the merits of our paper in your view.

---

> ### Comment · Reviewer_KReZ · 2024-08-13
>
> Thank you to the reviewers for the clarifications. I will stay moderately positive on this paper, and keep my score.

---

### Official Review · Reviewer_8sRE · 2024-07-17

**Soundness:** 3
**Presentation:** 3
**Contribution:** 3
**Rating:** 7
**Confidence:** 3

**Summary:**

.The paper studies DP-SGD under the assumption that the gradients have heavy-tailed phenomenon. This has been recently motivated and studied a lot by recent works. The authors claim to achieve optimal rate for this problem.

**Strengths:**

They obtain the first optimal rates (up to logarithmic factors) in the heavy-tailed setting, achieving error that depends on $\frac{G_1}{\sqrt{n}} + G_k(\sqrt{d}/nk)^{1-1/k}$ under approximate-DP guarantee.

They additionally study this problem under well-studied assumptions: Lipschitz constant assumption, smooth convex functions, and smooth generalized linear model.

I haven't checked the proof, but the paper is definitely an accept considering it solves a problem in this domain.

**Weaknesses:**

N/A

**Questions:**

N/A

---

> ### Author Rebuttal · Authors · 2024-08-06
>
> Thank you for your reviewing efforts. We appreciate your positive feedback. Please feel free to let us know if you have any questions or suggestions.

---

### Official Review · Reviewer_S4Vx · 2024-07-17

**Soundness:** 4
**Presentation:** 4
**Contribution:** 3
**Rating:** 7
**Confidence:** 2

**Summary:**

This paper gives three main results.

The first is a nearly optimal (losing a few logarithmic factors) excess loss rate for differentially private stochastic convex optimization (DP-SCO) when the gradient norms have $k$ bounded moments. In particular, they achieve the optimal excess loss under $\rho$-concentrated differential privacy (CDP) of

$\frac{G_2 D}{\sqrt{n}} + G_kD \cdot \left(\frac{\sqrt{d}}{n\sqrt{\rho}} \right)^{1-1/k}$

where $G_i$ denotes the uniform upper bound on the $i$th moment of the gradients and $D$ is the diameter of the domain. This improves over the prior state of the art [LR23] by a factor of $G_kD \cdot \left(\frac{\sqrt{d}}{n\sqrt{\rho}} \right)^{1/k}$. This improvement is increasingly pronounced as we decrease the number of gradient moments that we assume are bounded, which is probably the important regime in this work.

As I understand, the main technical innovation is a "population-level localization framework", through which the authors are able to control the population excess risk without having to argue about the variance of the bias term arising from gradient clipping.

The second main result is an algorithm that achieves the optimal excess risk when each sample function arrives with a known Lipschitz constant. This yields algorithms for privately learning a generalized linear model with the optimal excess risk. This follows from a clean reduction to the case where we know that our losses are uniformly Lipschitz.

The third main result is an improved query complexity for optimizing smooth functions. The algorithm follows from an application of the sparse vector technique.

**Strengths:**

The algorithms seem very natural, the analysis looks clean, and the results yield quantitative improvements over prior work. Therefore, I think this work is an important contribution to private convex optimization.

The ideas are cleanly explained and at least on a surface level make sense to a complete outsider to the field (such as myself).

**Weaknesses:**

Can't really think of anything significant. A natural criticism I anticipate is that the improvement over [LR23] is pretty minor when $k$ is large, but given that the authors get the optimal result and [LR23] doesn't, this doesn't seem like a real issue.

**Questions:**

N/A

**Limitations:**

Yes

---

> ### Author Rebuttal · Authors · 2024-08-06
>
> Thank you for your reviewing efforts, and we appreciate your positive feedback. Please feel free to let us know if you have any questions or suggestions.

---

### Official Review · Reviewer_UUcG · 2024-07-19

**Soundness:** 2
**Presentation:** 1
**Contribution:** 2
**Rating:** 3
**Confidence:** 4

**Summary:**

This paper studies the problem of differentially private stochastic convex optimization with heavy-tailed gradients. This paper points out that in typical optimization research, the assumption of uniformly G-Lipschitz, while convenient for bounding sensitivity, does not always hold. Based on this weakness, the authors studied k-heavy-tailed DP-SCO. The author obtains near-optimal algorithms that lie in the clipped DP-SGD subroutine and ensure the private minimization of a regularized ERM problem under k-heavy-tailed DP-SCO conditions. This method yields points that closely approximate the minimizer of population loss, verified through Markov’s inequality.

Updates post response:

- I find the lack of experiments a bit underwhelming, and the authors' response didn't really convince me why this should not be part of this paper. Data privacy is a highly practical question that should be of strong interest to practitioners. The lack of practical implications is thus concerning.

- The paper does not have a conclusion section, which is very strange for a NeurIPS paper. I have been reviewing NeurIPS submissions for many years; It is really awkward for a paper to not have a conclusions section. Not having a conclusions section also means that the authors did not carefully think about the "Limitations" of their work, which is a requirement as per NeurIPS submission guidelines.

- Lastly, I wonder what are some of the broader impacts/implications of this work. It would be helpful if the authors could add some discussions about this. Again, I feel like this would require a proper "conclusions" section added to the paper.

- While reading through the paper I noticed many notational inconsistencies and concepts that have not been clearly explained (please see concrete examples under the list of questions below), I suggest the authors to carefully proof read their paper and making improvements on explaining their work. This will help broaden the impact of this research.

Disclaimer: Please be aware that due to the rather short reviewing time, I have not been able to check all the proof details in the appendix.

**Strengths:**

- This paper provides a comprehensive analysis of prior work, enabling readers to quickly understand the limitations of previous research on DP-SCO and why the authors believe this work is necessary.
- This paper conducts solid theoretical research and has a profound understanding of the k-heavy-tailed DP-SCO problem. The entire paper is filled with rigorous derivations, providing a theoretical foundation for future research.

**Weaknesses:**

- Experiments could be added to validate the proposed algorithm for readers to better understand the key takeaways.

- The paper's organization could be improved, and the contribution can be better highlighted.

**Questions:**

- Why is the k-heavy-tailed assumption more common than the uniformly G-Lipschitz assumption? Intuitively, the uniform is indeed a weaker assumption, but I would like more direct evidence to prove the necessity of using the k-heavy-tailed assumption.

- What are the comparisons between the bounds under the k-heavy-tailed assumption with those of prior work? Please comment on this

- Is it possible to provide the convergence analysis of the proposed algorithm?

- From equation (1) to equation (2), what's the difference between $G$ and $G_2$?

- In line 45, can you give a specific example of what you mean by "heavy-tailed gradients"? When might readers expect this condition to hold in practice (e.g., in the training of a neural network)?

- While citing a paper, can you please properly cite the author's name(s) using \cite{}, as opposed to things like [LR23]?

- In the introduction, can you please clarify exactly under what notion of DP your results will hold? Note: I'm seeing multiple versions of DP definitions in section 2, page 5, thus prompting this question.

- Is Theorem 1 (at the bottom of page 9) supposed the main result in your paper?

- What are some of the broader impacts/implications of your work? Please comment on this.

**Limitations:**

Limitations are discussed on page 32.

---

> ### Author Rebuttal · Authors · 2024-08-06
>
> Thanks for your valuable feedback. We answer your questions below.
>
>
> Experiments: The focus of our work is theoretical in nature: developing algorithms that address known gaps in the literature on DP-SCO, a problem of interest in both theory and practice. We believe this goal has significant merit in its own right, as it provides insights that lay the groundwork for future research, both theoretical and experimental. We absolutely agree with the reviewer on the complementary importance of experimental evaluation of theoretical derivations. We chose to restrict our focus, but think that transferring our insights to practice is an exciting and promising direction for future research.
>
> Organization: We take the organization of our paper and presentation of our contributions seriously as concerns, and appreciate the feedback. We would like to kindly request that the reviewer provide some more specific feedback on our organization and presentation. Re: our contributions, all of our key results are described in Section 1.1 (with corresponding formal statements as Theorems 1-4), and compared to prior work in Section 1.2.
>
> Q1: Indeed, the k-heavy-tailed assumption applies to more situations than the uniform Lipschitz assumption, as a distribution can be $k$-heavy-tailed but not $\infty$-heavy-tailed. However, there is a tradeoff in the gains achievable for finite $k$. Our work is the first to obtain an algorithm that gives the best possible tradeoff up to logarithmic factors.
>
> In practice, differentially private neural networks are trained with clipped gradients (see [ACG+16]), due to outliers which create undesirable tail behavior. For more problems in practice where the $k$-heavy-tailed assumption is a more appropriate model than the uniform Lipschitz assumption, we refer the reviewer to the reference [WXDX20], which discusses how biomedicinal and financial research often use heavy-tailed modeling. More generally, power laws are heavy-tailed, and used to model many phenomena in the sciences. We will add additional discussion of this motivation; thank you for raising this point.
>
> Regarding the relative strength of these assumptions, for $k’ > k$, the $k’$-heavy-tailed assumption is stronger than the $k$-heavy-tailed assumption (with appropriate parameters), as a variable can have a finite $k$-th moment but not a finite $k'$-th moment. In this sense, as uniform Lipschitzness is an $\infty$-heavy-tailed assumption, it is the strongest.
>
> Q2: We compared our bounds to existing work in Lines 57-61 of the submission, as well as more in depth in Section 1.3. The heavy-tailed DP-SCO problem is relatively new, and few theoretical guarantees are provided by the literature. The two most relevant works are of [KLL22] and [LR23]: the first requires stringent conditions (such as uniformly smooth loss functions) to guarantee optimality, whereas the former obtains sub-optimal rates that can be polynomially worse than ours. We close the gap left by these prior works, achieving the first near-optimal rate, as well as generalizations in many settings (smooth and/or GLM).
>
> We were not sure if the reviewer meant how our Assumption 1 compared to previous definitions of the problem. Our problem statement is the same as in [LR23]. The problem definition is phrased slightly differently in [WXDX20, KLL22], using a (in our opinion, less natural) coordinatewise bound; we chose the definition consistent with more of the literature, e.g. previous works on private mean estimation [BD14], and uniform Lipschitz DP-SCO.
>
> Q3: We believe all of our algorithms have complete analyses and proofs. For example, Theorem 1 is proven in Section 3 and Appendix A, with a fully specified algorithm description. We are happy to address your concerns, but we would like to again kindly request that the reviewer be specific about which missing convergence analyses they refer to.
>
> We appreciate that you found our research solid and our descriptions comprehensive, providing a good foundation for the problem. We hope our response was clarifying, and that it elevates our contributions from your viewpoint. Thank you for your feedback again.

---

### Decision · Program_Chairs · 2024-09-25

**Decision:**

Accept (poster)

**Comment:**

This work studies differentially-private stochastic convex optimization (DP-SCO) under heavy-tailed (stochastic gradient) settings. Its main results are in closing the gap between upper and lower bounds for this problem. At the technical level the work puts forth some nonstandard ideas in this literature, such as using a population-localization approach (whereas it is far more typical to use empirical localization which is also simpler), and SVT as a privacy budget control under clipping. There were some concerns about the paper not containing a conclusions section or numerical experiments.

Overall, I think this paper makes enough contributions and merit to warrant publication at NeurIPS.